# Computing Optimal Nash Equilibria in Multiplayer Games

**Youzhi Zhang**
Centre for Artificial Intelligence and Robotics
Hong Kong Institute of Science & Innovation
Chinese Academy of Sciences
`youzhi.zhang@cair-cas.org.hk`

**Bo An**
School of Computer Science and Engineering
Nanyang Technological University
Singnapore
`boan@ntu.edu.sg`

**V. S. Subrahmanian**
Department of Computer Science
Northwestern University
Evanston, USA
`vss@northwestern.edu`

## Abstract

Designing efficient algorithms to compute a Nash Equilibrium (NE) in multiplayer games is still an open challenge. In this paper, we focus on computing an NE that optimizes a given objective function. For example, when there is a team of players independently playing against an adversary in a game (e.g., several groups in a forest trying to interdict illegal loggers in green security games), these team members may need to find an NE minimizing the adversary's utility. Finding an optimal NE in multiplayer games can be formulated as a mixed-integer bilinear program by introducing auxiliary variables to represent bilinear terms, leading to a huge number of bilinear terms, making it hard to solve. To overcome this challenge, we first propose a general framework for this formulation based on a set of correlation plans. We then develop a novel algorithm called **CRM** based on this framework, which uses **C**orrelation plans with their **R**elations to restrict the feasible solution space after the convex relaxation of bilinear terms while **M**inimizing the number of correlation plans to reduce the number of bilinear terms. We show that our techniques can significantly reduce the time complexity, and CRM can be several orders of magnitude faster than the state-of-the-art baseline.

## 1 Introduction

One of the important problems in artificial intelligence is the design of algorithms for agents to make decisions in interactive environments [33]. To this day, many results have been achieved in two-player non-cooperative environments, for example, security games [39], the game of Go [38], and poker games [6]. One of the most important solution concepts behind these results is the well-known Nash Equilibrium (NE) [30]. Indeed, there are many efficient algorithms, e.g., algorithms based on linear programs [42, 43, 37, 47, 48] or counterfactual regret minimization [56, 5], to compute Nash equilibria (NEs) in two-player zero-sum games. However, there are fewer results on efficient algorithms for NEs with theoretical guarantees in multiplayer games (see the discussion in [4]), and most of these results are for games with particular structures (e.g., polymatrix games [7, 10]). The main reason is that finding NEs in multiplayer games is hard — it is PPAD-complete even for zero-sum three-player games [8]. Designing efficient algorithms to compute NEs in multiplayer games is thus still an open challenge.

37th Conference on Neural Information Processing Systems (NeurIPS 2023).

In this paper, we focus on computing an optimal NE that optimizes a specific objective over the space of NEs. In the real world, we may need to optimize our objective over the space of NEs [34]. Possible objectives [9] could be maximizing social welfare (the sum of the players' expected utilities), maximizing the expected utilities of one player or several players, maximizing the minimum utility among players, minimizing the support sizes of the NE strategies, and so on. In addition, when there is a team of players in a game, team members need to consider finding an equilibrium that optimizes some objective [45, 12]. For example, in green security games where several heterogeneous groups (e.g., local police, the Madagascar National Parks, NGOs, and community volunteers) try to protect forests from illegal logging [26], the groups involved may need to find an NE that minimizes the adversary's utility[1].

Unfortunately, the problems mentioned above are NP-hard [14, 9]. In two-player games, finding an optimal NE can be formulated as a mixed-integer linear program [34]. In this formulation, finding an optimal solution means optimizing an objective over the space of NEs, and this space is modeled as the feasible solution space of the mixed-integer linear program. We can directly extend this two-player formulation to find an optimal NE in multiplayer games by representing the space of NEs as the feasible solution space of a mixed-integer bilinear program transformed from a multilinear program by using auxiliary variables to represent bilinear terms. Then finding an optimal NE requires solving a non-convex program. Unfortunately, such a formulation is not efficient because there are exponentially many bilinear terms in the program. There are other approaches (e.g., [4]) that guarantee finding an NE in multiplayer games. However, these approaches need to enumerate all NEs to find an optimal NE, which is very inefficient [34] (see our experimental results) because there can be exponentially many NEs [44].

To tackle this challenge, we first propose a general framework for transforming a multilinear program for computing optimal NEs into a bilinear program based on a set of correlation plans, where each correlation plan (i.e., a probability distribution over joint actions) corresponds to a set of auxiliary variables representing a set of bilinear terms. We then develop a novel algorithm called **CRM** based on this framework, which uses **C**orrelation plans with their **R**elations to strictly reduce the feasible solution space after the convex relaxation of bilinear terms while **M**inimizing the number of correlation plans to reduce the number of bilinear terms. We show that our techniques can significantly reduce the time complexity, and CRM can be several orders of magnitude faster than the state-of-the-art baseline. To our best knowledge, CRM is the first algorithm to use a minimum set of correlation plans to reformulate the program for computing optimal NEs in multiplayer games.

## 2   Preliminaries

Consider a normal-form game[2] $G = (N, A, u)$ [37]. We denote the set of players as $N = \{1, \ldots, n\}$; the set of all players' joint actions is $A = \times_{i \in N} A_i$, where $A_i$ is the finite set of player $i$'s pure strategies (actions) with $a_i \in A_i$; and the set of all players' payoff functions is $u = (u_1, \ldots, u_n)$, where $u_i : A \to \mathbb{R}$ is player $i$'s payoff function. Let $U_{max} = \max_{i \in N} \max_{a \in A} u_i(a)$, and $U_{min} = \min_{i \in N} \min_{a \in A} u_i(a)$. In addition, the set of (joint) mixed strategy profiles $X = \times_{i \in N} X_i$, where $X_i = \Delta(A_i)$ (i.e., the set of probability distributions over $A_i$) is the set of mixed strategies of player $i$, and $x_i(a_i)$ is the probability that any action $a_i \in A_i$ is played. Let $-i$ be the set of all players excluding player $i$, i.e., $-i = N \setminus \{i\}$, and $A_{-i}$ be $\times_{j \in N \setminus \{i\}} A_j$. Generally, given $N' \subseteq N$, $a_{N'} \in A_{N'} = \times_{i \in N'} A_i$, $a_{N'}(i)$ is the action of player $i \in N'$ in the joint action $a_{N'}$. For example, if $a_{N'} = (a_1, a_3, a_5)$ with $N' = \{1, 3, 5\}$, $a_{N'}(3) = a_3$. If $N' = N$, we ignore the subscript, i.e., $a = a_N$ and $A = A_N$. For each $x \in X$, player $i$'s expected payoff is:

$$u_i(x) = \sum_{a \in A} u_i(a) \prod_{j \in N} x_j(a(j)),$$

and, if player $i$ plays $a_i$:

$$u_i(a_i, x_{-i}) = \sum_{a_{-i} \in A_{-i}} u_i(a_i, a_{-i}) \prod_{j \in -i} x_j(a_{-i}(j)).$$

---

[1]Here, if all team members play strategies according to an NE minimizing the adversary's utility, the adversary cannot deviate from the equilibrium strategy to obtain a higher utility.

[2]Our methods mostly apply to normal-form games including green security games mentioned in Section 1. Extensive-form games can be first converted to normal-form games to be solved, and exploiting their game structure is the future work.

In this paper, we consider multiplayer games, i.e., $n > 2$.

A Nash Equilibrium (NE, and NEs for Nash Equilibria) [30] is a stable strategy profile in which no player has an incentive to change her strategy given other players' strategies and always exists. Formally, a strategy profile $x^*$ is an NE if, for each player $i$, $x_i^*$ is a best response to $x_{-i}^*$, i.e., $u_i(x_i^*, x_{-i}^*) \geq u_i(x_i, x_{-i}^*), \forall x_i \in X_i$, which is equivalent to $u_i(x_i^*, x_{-i}^*) \geq u_i(a_i, x_{-i}^*), \forall a_i \in A_i$.

With the above condition of NEs, we could use a multilinear program to represent the space of NEs, but it will involve the product of strategies in $u_i(x)$, whose degree is $n$ and is higher than the product of strategies in $u_i(a_i, x_{-i})$. To reduce the degree of the program representing the space of NEs from $n$ to $n-1$ (i.e., only the product of strategies in $u_i(a_i, x_{-i})$ is required), in two-player games, the previous work [34] exploited the following NE's property, which can be used in multiplayer games as well. For each strategy profile $x \in X$, the regret of an action $a_i$ is the difference in player $i$'s expected utility between playing $x_i$ in $x$ and playing $a_i$, i.e., $u_i(x) - u_i(a_i, x_{-i})$. Obviously, a strategy profile $x \in X$ is an NE if and only if every action either has the regret 0, or is played with the probability 0 in $x$. Then the space of NEs of a game can be formulated as the feasible solution space of a mixed-integer program by using a binary variable $b_{a_i}$ to represent that any action $a_i$ either has the regret 0, or is played with the probability 0:

$$u_i(a_i, x_{-i}) = \sum_{a_{-i} \in A_{-i}} u_i(a_i, a_{-i}) \prod_{j \in -i} x_j(a_{-i}(j)) \quad \forall i, a_i \in A_i \tag{1a}$$

$$\sum_{a_i \in A_i} x_i(a_i) = 1 \quad \forall i \in N \tag{1b}$$

$$1 - b_{a_i} \geq x_i(a_i) \quad \forall i \in N, a_i \in A_i \tag{1c}$$

$$u_i(x) \geq u_i(a_i, x_{-i}) \quad \forall i \in N, a_i \in A_i \tag{1d}$$

$$u_i(x) - u_i(a_i, x_{-i}) \leq b_{a_i}(U_{max} - U_{min}) \quad \forall i \in N, a_i \in A_i, \tag{1e}$$

$$u_i(a_i, x_{-i}) \in [U_{min}, U_{max}], u_i(x) \in [U_{min}, U_{max}] \quad \forall i \in N, a_i \in A_i, \tag{1f}$$

$$x_i(a_i) \in [0, 1], b_{a_i} \in \{0, 1\}, \quad \forall i \in N, a_i \in A_i, \tag{1g}$$

where we use the notations of utility functions $u_i(x)$ and $u_i(a_i, x_{-i})$ to represent the corresponding variables in the program. Eq.(1c) ensures that binary variable $b_{a_i}$ is set to 0 when $x_i(a_i) > 0$ and can be set to 1 only when $x_i(a_i) = 0$; and Eq.(1e) ensures that the regret of action $a_i$ equals 0 (i.e., $u_i(x) = u_i(a_i, x_{-i})$), unless $b_{a_i} = 1$ where the constraint $u_i(x) - u_i(a_i, x_{-i}) \leq (U_{max} - U_{min})$ always holds.

An optimal NE is an NE optimizing an objective function $g(x)$ over the space of NEs, where $g(x)$ is a linear objective function[3] and could be maximizing social welfare, maximizing the expected utilities of one player or several players, maximizing the minimum utility among players, minimizing the support sizes of the NE strategies, and so on. Unfortunately, finding an optimal NE optimizing the above objectives is NP-hard [9].

## 3   Computing Optimal Nash Equilibria

The problem of finding an optimal NE in multiplayer games requires optimizing an objective over the space of NEs. This space is represented by Eq.(1), which involves nonlinear terms in Eq.(1a) to represent the strategies of players in $-i$, which is bilinear when $n = 3$ and is multilinear when $n \geq 4$. The multilinear program is usually transformed into a bilinear program to make the program solvable using global optimization solvers, e.g., Gurobi [19]. Here, we propose a general framework for this transformation based on a set of correlation plans for any binary collection of subsets of players, where each set in this collection is divided into two disjoint sets, and each correlation plan corresponds to a set of auxiliary variables representing a set of bilinear terms. However, there are two challenges for solving this bilinear program: 1) this bilinear program usually involves a large number of bilinear terms, and 2) an important step used by state-of-the-art algorithms to solve such bilinear programs is to use convex relaxation to replace each bilinear term in the program [15, 18], which significantly enlarges the feasible solution space. To overcome these challenges, we develop a novel algorithm called **CRM** that uses **C**orrelation plans with their **R**elations to strictly reduce the feasible

---

[3]If $g(x)$ is nonlinear, we can use a variable (i.e., a linear function) $v$ as the new objective with the constraint such that $v = g(x)$.

solution space after the convex relaxation while **M**inimizing the number of correlation plans to reduce the number of bilinear terms. Section 3.4 shows that our techniques can significantly reduce the time complexity. The procedure of CRM is shown in Algorithm 2, which is illustrated in Appendix A.

### 3.1 A General Transformation Framework

A correlation plan is a probability distribution over the joint action space of a subset of players, and we focus on correlation plans for certain special collections of subsets of players, which can be used to transform a multilinear program for computing optimal NEs into a bilinear program.

**Definition 1.** *A collection $\mathcal{N}$ of subsets of players is a **binary collection** if:*

1. *$\{-i \mid i \in N\} \subseteq \mathcal{N}$;*

2. *for each $N' \in \mathcal{N}$, $N' \subset N$ with $|N'| \geq 2$; and*

3. *for each $N' \in \mathcal{N}$, there are two disjoint children $N'_l$ and $N'_r$ in $\{\{i\} \mid i \in N\} \cup \mathcal{N}$ such that $N'_l \cap N'_r = \emptyset$ and $N' = N'_l \cup N'_r$, i.e., $N'$ is divided into two disjoint sets.*

*Let $N'_l$ and $N'_r$ be the left child and the right child of $N' \in \mathcal{N}$, respectively. For each $N'$ in any binary collection $\mathcal{N}$, a **correlation plan** of $N'$ is a probability distribution $x_{N'}$ over $A_{N'}$: given $x_{N'}(a_{N'}) \in [0,1]$ ($\forall a_{N'} \in A_{N'}, N' \in \mathcal{N}$),*

$$\sum_{a_{N'} \in A_{N'}} x_{N'}(a_{N'}) = 1 \quad \forall N' \in \mathcal{N}. \tag{2}$$

For simplification, let $i$ be equivalent to $\{i\}$ for each $i \in N$. That is, $x_i$ is a special correlation plan $x_{\{i\}}$ (i.e., $x_i = x_{\{i\}}$), $a_i \in A_i$ is a special joint action $a_{\{i\}} \in A_{\{i\}}$ (i.e., $a_i = a_{\{i\}}$). Each element $N'$ in a binary collection $\mathcal{N}$ has the binary division, i.e., it is divided into two disjoint sets $N'_l$ and $N'_r$. Based on this binary division, any joint action $a_{N'} \in A_{N'}$ can be divided into two sub-joint actions $a_{N'_l} \in A_{N'_l}$ and $a_{N'_r} \in A_{N'_r}$ such that $a_{N'} = (a_{N'_l}, a_{N'_r})$. Then we can use this binary division to ensure that $\prod_{j \in N'} x_j(a_{N'}(j)) = x_{N'}(a_{N'})$ for the correlation plan $x_{N'}$, as shown in Example 1.

**Example 1.** $\{\{1,2,3\}, \{1,2,4\}, \{1,3,4\}, \{2,3,4\}, \{1,2\}, \{1,3\}, \{2,3\}, \{1,4\}, \{3,4\}, \{2,4\}\}$ *is a binary collection for a four-player game. For $N' = \{1,2,3\}$ in this collection with $N'_l = \{1,2\}$ (having two children $\{1\}$ and $\{2\}$) and $N'_r = \{3\}$, we have $a_{N'} = (a_1, a_2, a_3) = (a_{\{1,2\}}, a_3) \in A_{N'}$ and $a_{\{1,2\}} = (a_1, a_2) \in A_{\{1,2\}}$. Then we can have a chain of bilinear constraints (equalities): $x_{N'}(a_{N'}) = x_{\{1,2\}}(a_{\{1,2\}})x_3(a_3)$ and $x_{\{1,2\}}(a_{\{1,2\}}) = x_1(a_1)x_2(a_2)$, which guarantees that $x_{N'}(a_{N'}) = x_1(a_1)x_2(a_2)x_3(a_3)$. In other words, we use $x_{\{1,2\}}(a_{\{1,2\}})$ and $x_{N'}(a_{N'})$ as the auxiliary variables to represent bilinear terms $x_1(a_1)x_2(a_2)$ and $x_{\{1,2\}}(a_{\{1,2\}})x_3(a_3)$, respectively.*

This property of correlation plans of a binary collection $\mathcal{N}$ can be used to transform the multilinear Program (1) into a bilinear program. First, we use the binary division of each element $N'$ in $\mathcal{N}$ to connect correlation plans, i.e., for any $N' \in \mathcal{N}$ with its children $N'_l$ and $N'_r$:

$$x_{N'}(a_{N'}) = x_{N'_l}(a_{N'_l})x_{N'_r}(a_{N'_r}) \quad \forall a_{N'} = (a_{N'_l}, a_{N'_r}) \in A_{N'} \tag{3a}$$

$$x_{N'}(a_{N'}) \in [0,1] \quad \forall a_{N'} \in A_{N'}. \tag{3b}$$

Second, we replace $\prod_{j \in -i} x_j(a_{-i}(j))$ in Eq.(1a) with $x_{-i}(a_{-i})$:

$$u_i(a_i, x_{-i}) = \sum_{a_{-i} \in A_{-i}} u_i(a_i, a_{-i})x_{-i}(a_{-i}) \quad \forall i \in N, a_i \in A_i. \tag{4}$$

In the above transformation, each correlation plan corresponds to a set of auxiliary variables (e.g., $x_{N'}(a_{N'})$) representing a set of bilinear terms (e.g., $x_{N'_l}(a_{N'_l})x_{N'_r}(a_{N'_r})$). Eq.(3) guarantees that $\prod_{j \in N'} x_j(a_{N'}(j)) = x_{N'}(a_{N'})$, and then the feasible solution space of Eqs.(1b)-(1g), (3), and (4) represents the space of NEs.

**Theorem 1.** *The feasible solution space of mixed strategies (i.e., $x_i(a_i)$ for each $i \in N$, $a_i \in A_i$) in Eqs.(1b)-(1g), (3), and (4) is the space of NEs. (Proofs are in Appendix B.)*

We can then compute an optimal NE by solving the following mixed-integer bilinear program according to any binary collection $\mathcal{N}$:

$$\max_x g(x) \tag{5a}$$

$$\text{s.t. Eqs.}(1b) - (1g), (3), (4). \tag{5b}$$

It is straightforward to solve Program (5) by using the **vanilla binary collection** $\overline{\mathcal{N}}$ that includes all non-singleton proper subsets of $N$, i.e., $\overline{\mathcal{N}} = \{N' \mid N' \subset N, |N'| \geq 2\}$, where, for each $N' \in \overline{\mathcal{N}}$, $N'_l$ is $N' \setminus \{j\}$ and $N'_r$ is $\{j = \max_{i \in N'} i\}$. Example 1 provides an example of $\overline{\mathcal{N}}$.

### 3.2 Exploit Correlation Plans with Their Relations

In this section, we use correlation plans with their relations to restrict the feasible solution space after the convex relaxation. The common convex relaxation technique [27, 35, 18] before searching for the optimal solution is: each bilinear term $x_{N'}(a_{N'}) = y_1 y_2$ with $y_1, y_2 \in [0, 1]$ is represented by the following constraints including four linear constraints:

$$\max\{0, y_1 + y_2 - 1\} \leq x_{N'}(a_{N'}) \leq \min\{y_1, y_2\}, \tag{6}$$

which significantly enlarges the feasible solution space. We now show the motivation to use correlation plans with their relations to reduce this feasible solution space.

**Example 2.** *Given $N' = \{2, 4\} \subset N$ with two actions for each player (i.e., $A_i = \{a_i, a'_i\}$) in a game $G$, by Eq.(6), bilinear terms (e.g., $x_{N'}(a_2, a_4) = x_2(a_2)x_4(a_4)$) are relaxed according to Eq.(6), e.g., $\max\{0, x_2(a_2) + x_4(a_4) - 1\} \leq x_{N'}(a_2, a_4) \leq \min\{x_2(a_2), x_4(a_4)\}$. With additional constraints by Eq.(1b) (e.g., $x_4(a_4) + x_4(a'_4) = 1$), the following assignment could be a feasible solution:*

$$x_{N'}(a_2, a_4) = x_{N'}(a'_2, a_4) = x_{N'}(a_2, a'_4) = x_{N'}(a'_2, a'_4)$$
$$= x_2(a_2) = x_2(a'_2) = x_4(a_4) = x_4(a'_4) = 0.5. \tag{7}$$

*Obviously, in Eq.(7), $x_{N'}(a_2, a_4)$ is not equal to $x_2(a_2)x_4(a_4)$. In fact, based on Eq.(2), we have:*

$$x_{N'}(a_2, a_4) + x_{N'}(a'_2, a_4) + x_{N'}(a_2, a'_4) + x_{N'}(a'_2, a'_4) = 1, \tag{8}$$

*which will make the solution in Eq.(7) infeasible. Moreover, the following assignment is a feasible solution after the relaxation and satisfies Eq.(8):*

$$x_{N'}(a_2, a_4) = x_{N'}(a'_2, a_4) = x_{N'}(a_2, a'_4) = 0.2,$$
$$x_{N'}(a'_2, a'_4) = 0.4, x_2(a_2) = x_4(a_4) = 0.5. \tag{9}$$

*However, in Eq.(9), $x_{N'}(a_2, a_4)$ is still not equal to $x_2(a_2)x_4(a_4)$. Actually, we have:*

$$x_{N'}(a_2, a_4) + x_{N'}(a'_2, a_4) = x_4(a_4),$$
$$x_{N'}(a_2, a'_4) + x_{N'}(a'_2, a'_4) = x_4(a'_4), \tag{10}$$

*which will make the solution in Eq.(9) infeasible.*

This above example shows that we can use the definition of a correlation plan (i.e., Eq.(2)) and the relation of correlation plans to reduce the feasible solution space after the relaxation.

Each element $N'$ in any binary collection $\mathcal{N}$ is defined by $N'_l$ and $N'_r$, which actually defines relations between correlation plans for elements in $\{\{i\} \mid i \in N\} \cup \mathcal{N}$. In Example 2, Eq.(10) actually represents a relation between the correlation plan $x_{N'}$ and the mixed strategy $x_4$ (i.e., the special correlation plan $x_{\{4\}}$). Formally, for any $N' \in \mathcal{N}$ and $i \in N'$:

$$\sum_{a_{N'} \in A_{N'}, a_{N'}(i) = a_i} x_{N'}(a_{N'}) = x_i(a_i) \quad \forall a_i \in A_i, \tag{11}$$

where $a_{N'}(i)$ is the action of player $i \in N'$ in the joint action $a_{N'}$. Similarly, let $a_{N'}(N'')$ be the sub-joint action of player $N'' \subset N'$ in the joint action $a_{N'}$. Then, for any $N' \in \mathcal{N}$ with its children $N'_l$ and $N'_r$, and $N'' \in \{N'_l, N'_r\}$:

$$\sum_{a_{N'} \in A_{N'}, a_{N'}(N'') = a_{N''}} x_{N'}(a_{N'}) = x_{N''}(a_{N''}) \quad \forall a_{N''} \in A_{N''} \tag{12}$$

---
**Algorithm 1** Generate $\underline{\mathcal{N}}$
---
1: Build a full binary tree $T_{-n}$ with the height $\lceil \log_2(n-1) \rceil$ for $-n$ with the set of internal nodes $\mathcal{N}_{T_{-n}}$ and $|\mathcal{N}_{T_{-n}}| = n - 2$
2: **for** each $i$ in $\{1, \ldots, n-1\}$ **do**
3:     Search $T_{-n}$ to replace $i$ with $n$ in each node including $i$ to form a binary tree $T_{-i}$ with the set of internal nodes $\mathcal{N}_{T_{-i}}$
4: **end for**
5: $\underline{\mathcal{N}} \leftarrow \cup_{i \in N} \mathcal{N}_{T_{-i}}$.
---

where we could add $|N''| > 1$ to ensure that Eq.(11) and Eq.(12) do not generate the same constraints. Appendix A shows some examples of these constraints. Eq.(11) represents the relation between the correlation plan $x_{N'}$ and the mixed strategy $x_i$ (i.e., the special correlation plan $x_{\{i\}}$) for each $i \in N'$, and Eq.(12) represents the relation between the correlation plan $x_{N'}$ and the correlation plan $x_{N'_l}$ or $x_{N'_r}$. Equivalently, Eqs.(2), (11), and (12) represent the marginalization constraints that independent probability distributions ought to obey, where $x_{N'}$ is the joint distribution (represented by Eq.(2)) of independent distributions $x_i$ for all $i \in N'$ (represented by Eq.(11)) or independent distributions $x_{N'_r}$ and $x_{N'_l}$ (represented by Eq.(12)).

We now show the effectiveness of our correlation plans by showing our method strictly reduces the feasible solution space after the relaxation. Reducing the feasible solution space will make the program efficiently solvable, as shown in the experiments. Let $\mathcal{M}$ be the original feasible solution space for the original multilinear program that is transformed into a bilinear program according to $\mathcal{N}$, i.e., $\mathcal{M}$ is particularly constrained by Eqs.(1b), (3a), and (3b). We define the convex relaxation space $\mathcal{R}$ as using Eq.(6) to represent each bilinear term in Eq.(3a), i.e., $\mathcal{R}$ is particularly constrained by Eqs.(1b), (6), and (3b). We define our tight relaxation space $\mathcal{T}$ based on our correlation plans, i.e., $\mathcal{T}$ is particularly constrained by Eqs.(1b), (2), (11), (12), and (3b). (Proofs are in Appendix B.)

**Theorem 2.** $\mathcal{M} \subset \mathcal{T} \subset \mathcal{R}$, *i.e.,* $\mathcal{T}$ *is strictly smaller than* $\mathcal{R}$ *but still includes* $\mathcal{M}$.

The property $\mathcal{M} \subset \mathcal{T} \subset \mathcal{R}$ means that: i) we can use $\mathcal{T}$ to strictly reduce the feasible solution space after the relaxation, and ii) restricting the feasible solution space to $\mathcal{T}$ does not reduce the space of NEs and then guarantees optimality for the original program. We now explicitly restrict the feasible solution space to $\mathcal{T}$ by adding Eqs.(2), (11), and (12) to Program (5) for any binary collection $\mathcal{N}$:

$$\max_x g(x) \tag{13a}$$

$$\text{s.t. Eqs.}(1b) - (1g), (2), (3), (4), (11), (12). \tag{13b}$$

**Theorem 3.** *The optimal solution of Program (13) maximizes* $g(x)$ *over the space of NEs.*

Using the bilinear constraint Eq.(3a) in Program (13) is necessary for computing an optimal NE by solving Program (13) because $\mathcal{M} \neq \mathcal{T}$. Appendix C shows that, after removing Eq.(3a) in Program (13), the inefficiency can be arbitrarily large, and the resulting strategy profile may not be an NE.

## 3.3 Minimum-Height Binary Trees

In Program (13), we need to add a set of linear constraints and bilinear constraints for each correlation plan corresponding to each element in any binary collection $\mathcal{N}$. The size of the vanilla binary collection $\overline{\mathcal{N}}$ is $2^n - (n+2)$, which grows exponentially with the number of players. In this section, we propose building minimum-height binary trees to obtain a minimum binary collection. Our binary collection gives us a minimum set of correlation plans, which requires significantly fewer bilinear terms than $\overline{\mathcal{N}}$.

There are different ways to divide a subset of players, which determines different binary collections. For example, $N' = \{1, 2, 3, 4\}$ can be divided into $\{1, 2\}$ and $\{3, 4\}$ or $\{1\}$ and $\{2, 3, 4\}$, which will lead to different binary collections. Therefore, for obtaining a minimum binary collection $\mathcal{N}$, the challenge is how to effectively divide each element in $\mathcal{N}$.

---

**Algorithm 2** CRM

1: **Input:** A game $G = (N, A, u)$ and an objective function $g(x)$
2: A binary collection $\underline{\mathcal{N}} \leftarrow$ The output of Algorithm 1
3: Create Eqs.(1b)-(1g)
4: Create Eqs.(3), (2), (4), (11), and (12) according to $\underline{\mathcal{N}}$
5: $x^* \leftarrow$ An optimal solution by solving Program (13) based on $\underline{\mathcal{N}}$, i.e., $\max_x g(x)$ s.t. Eqs.$(1b) -$ $(1g), (2), (3), (4), (11), (12)$
6: **Output:** An optimal NE $x^*$.

---

To overcome this challenge, we propose building a minimum-height binary tree for each element in $\{-i \mid i \in N\}$ and ensuring that the number of internal nodes in these binary trees is the minimum. The binary division for each element in a binary collection $\mathcal{N}$ creates a binary tree for each

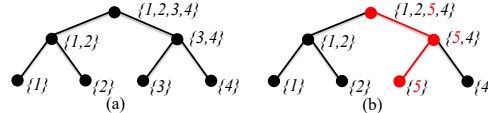

Figure 1: Binary trees for $-5$ and $-3$

element in $\{-i \mid i \in N\}$. For example, Figure 1(a) is a binary tree for $-5 = \{1, 2, 3, 4\}$, and Figure 1(b) is a binary tree for $-3 = \{1, 2, 5, 4\}$ in five-player games. Each binary tree for $-i$ is a full binary tree, i.e., each internal node has two children, with $n - 2$ internal nodes and $n - 1$ leaf nodes, where the height is the number of internal nodes on the longest path from the root to a leaf (e.g., the height in Figure 1(a) is 2). Details for these binary trees are shown in Appendix D. We can then build a full binary tree $T_{-n}$ with the minimum height $\lceil \log_2(n - 1) \rceil$ for $-n$ and then replace $i$ with $n$ in the nodes of $T_{-n}$ to obtain $T_{-i}$ for each $i \in -n = \{1, \ldots, n - 1\}$. That creates $n$ full binary trees for $\{-i \mid i \in N\}$. This procedure is shown in Algorithm 1 (details are shown in Appendix D), generating our **minimum binary collection** $\underline{\mathcal{N}}$ including all internal nodes in these trees. For example, Figure 1(a) builds a binary tree $T_{-5}$, and Figure 1(b) obtains $T_{-3}$ by replacing 3 with 5 in $T_{-5}$. Generally, we only need to create at most $\lceil \log_2(n - 1) \rceil$ new internal nodes to build a minimum-height binary tree for each $-i$ with $i \in -n$. Then $|\underline{\mathcal{N}}|$ is at most $n - 2 + (n - 1)\lceil \log_2(n - 1) \rceil$, i.e., $O(n \log n)$.

**Theorem 4.** $\underline{\mathcal{N}}$ *generated by Algorithm 1 is a binary collection, and $O(n \log n)$ for the size of $\underline{\mathcal{N}}$ is the minimum size of all binary collections of a game $G$. (Proofs are in Appendix B.)*

$|\underline{\mathcal{N}}|$ only grows sub-quadratically with $n$ and is much smaller than $|\overline{\mathcal{N}}| = 2^n - (n + 2)$ for $\overline{\mathcal{N}}$. Then $\underline{\mathcal{N}}$ requires fewer bilinear terms than $\overline{\mathcal{N}}$ when $n > 3$. For example, in a seven-player game with two actions for each player, by using $\underline{\mathcal{N}}$ with $|\underline{\mathcal{N}}| = 21$ correlation plans, the number of bilinear terms is 564, which is much smaller than 2044 by using $\overline{\mathcal{N}}$ with $|\overline{\mathcal{N}}| = 119$ correlation plans. Table 3 of Appendix G shows more examples. Note that Algorithm 1 cannot reduce the number of internal nodes when $n = 3$ because each element in $\{-i \mid i \in N\}$ includes only two players in three-player games. Our algorithm, CRM, is solving Program (13) based on $\underline{\mathcal{N}}$, which is shown in Algorithm 2 and is illustrated in Appendix A.

### 3.4 Complexity

The problem of finding an optimal NE is NP-hard [9], and our algorithm, CRM, i.e., Program (13) based on $\underline{\mathcal{N}}$ generated by Algorithm 1, is a mixed-integer bilinear program, whose scalability is mainly affected by the number of bilinear terms and integer variables. Generally, the problem of solving a linear integer program is NP-hard, and the time complexity is $O(I^2 (EC^2)^{2E+3})$ [32], where $I$ is the number of integer variables, $E$ is the number of constraints containing integer variables, and $C$ is the maximum value among constants and the range of integer variables in these constraints. Theoretically, each bilinear term can be represented by a mixed-integer linear program by introducing a new set of constraints and binary integer variables [21]. Suppose each bilinear term introduces $I'$ integer variables and $E'$ constraints, and each player has $m$ actions. Program (5) based on $\overline{\mathcal{N}}$ has $mn$ binary integer variables with $mn$ constraints and the following number of bilinear terms:

$$\sum_{N' \in \overline{\mathcal{N}}} \prod_{i \in N'} |A_i| \le (2^n - n - 2)m^{n-1} \le 2^n m^{n-1}.$$

Then the time complexity for solving the Program (5) based on $\overline{\mathcal{N}}$ is $O(I_1^2 (E_1 C^2)^{2E_1+3})$ where $I_1 = 2^n m^{n-1} I' + mn$ and $E_1 = 2^n m^{n-1} E' + mn$. Program (13) based on $\underline{\mathcal{N}}$ has $mn$ binary

Table 1: Part of experimental results (more results are in Tables 4 and 5 of Appendix H). The format is: Average Runtime $\pm$ 95% Confidence Interval (Percentage of Games not Solved within the Time Limit) (Utility Gap). Note that the unit of the runtime is second, the case that all games have been solved with the time limit should be $(0\%)$ and is omitted, we only need to care about the utility gap (a larger gap means losing more) for EXCLUSION, and the utility gap $\infty$ represents EXCLUSION cannot return a solution within the time limit.

| Random Game | | Runtime $\pm$ 95% Confidence Interval (Percentage of Games not Solved) (Utility Gap) | | | |
|---|---|---|---|---|---|
| Vary | $(n,m)$ | CRM | MIBP | ENUMPOLY | EXCLUSION |
| | (3, 2) | **0.01 $\pm$ 0** | 0.02 $\pm$ 0 | 0.03 $\pm$ 0.01 | 31 $\pm$ 41 (gap:15%) |
| $n$ | (5, 2) | **0.2 $\pm$ 0.1** | 0.5 $\pm$ 0.4 | 11 $\pm$ 4 | 753 $\pm$ 148 (73%) (gap:64%) |
| | (7, 2) | **25 $\pm$ 17** | 429 $\pm$ 131 (20%) | 1000 $\pm$ 0 (97%) | 835 $\pm$ 119 (80%) (gap:53%) |
| | (3, 5) | **0.2 $\pm$ 0.03** | 0.3 $\pm$ 0.1 | 1000 $\pm$ 0 (100%) | 1000 $\pm$ 0 (100%) (gap:67%) |
| $m$ | (3, 8) | **4 $\pm$ 3** | 247 $\pm$ 140 (17%) | 1000 $\pm$ 0 (100%) | 1000 $\pm$ 0 (100%) (gap:$\infty$) |
| | (3, 10) | **9 $\pm$ 9** | 334 $\pm$ 167 (30%) | 1000 $\pm$ 0 (100%) | 1000 $\pm$ 0 (100%) (gap:$\infty$) |
| | (3, 13) | **38 $\pm$ 21** | 342 $\pm$ 151 (27%) | 1000 $\pm$ 0 (100%) | 1000 $\pm$ 0 (100%) (gap:$\infty$) |
| GAMUT Game | | CRM | MIBP | ENUMPOLY | EXCLUSION |
| Random LEG | | **2 $\pm$ 1** | 1000 $\pm$ 0 (100%) | 1000 $\pm$ 0 (100%) | 986 $\pm$ 27 (97%) (gap:11%) |
| Random graphical | | **0.1 $\pm$ 0.1** | 803 $\pm$ 140 (83%) | 50 $\pm$ 30 | 971 $\pm$ 55 (97%) (gap:32%) |
| Uniform LEG | | **2.2 $\pm$ 1** | 1000 $\pm$ 0 (100%) | 1000 $\pm$ 0 (100%) | 986 $\pm$ 26 (97%) (gap:11%) |

Table 2: Ablation study (more results are in Table 6 of Appendix H). Note that $\overline{\mathcal{N}}$ (in CR and C) and $\underline{\mathcal{N}}$ (in CRM, CM, and M) result in the same bilinear terms in three-player games because each element in $\{-i \mid i \in \{1, 2, 3\}\}$ includes only two players such that Algorithm 1 cannot reduce the number of internal nodes to reduce the number of bilinear terms, and then CR and CRM (or C and CM) have the same performance. The unit of the runtime is second.

| | Runtime $\pm$ 95% Confidence Interval (Percentage of Games not Solved) | | | | |
|---|---|---|---|---|---|
| Game | CRM | CR | CM | C | M |
| (8, 2) | **156$\pm$ 83 (3%)** | 612$\pm$ 129 (33%) | 190 $\pm$ 102 (7%) | 763 $\pm$ 120 (60%) | 1000 $\pm$ 0 (100%) |
| (7, 2) | **25 $\pm$ 17** | 89 $\pm$ 51 | 36 $\pm$ 28 | 408 $\pm$ 157 (30%) | 488 $\pm$ 111 (10%) |
| (3, 15) | **167$\pm$ 86 (3%)** | 167 $\pm$ 86 (3%) | 317 $\pm$ 137 (17%) | 317 $\pm$ 137 (17%) | 558 $\pm$ 150 (40%) |
| (3, 17) | **231$\pm$122 (10%)** | 231 $\pm$122 (10%) | 326 $\pm$ 134 (20%) | 326 $\pm$ 134 (20%) | 784 $\pm$ 102 (53%) |
| Random graphical | **0.1 $\pm$ 0.1** | 0.4 $\pm$ 0.1 | 0.2 $\pm$ 0.1 | 0.6 $\pm$ 0.4 | 814 $\pm$ 134 (80%) |
| Uniform LEG | **2.2 $\pm$ 1** | 5 $\pm$ 4 | 2.5 $\pm$ 2 | 5 $\pm$ 5 | 999 $\pm$ 2 (97%) |

integer variables with $mn$ constraints and the following number of biliear terms:

$$\sum_{N' \in \underline{\mathcal{N}}} \prod_{i \in N'} |A_i| \leq |\underline{\mathcal{N}}| m^{n-1},$$

i.e., $O((n \log n)m^{n-1})$ bilinear terms (this size is the minimum because $O(n \log n)$ is the minimum size of binary collections by Theorem 4) and . Then the time complexity for solving Program (13) based on $\underline{\mathcal{N}}$ is $O(I_2^2(E_2 C^2)^{2E_2+3})$ where $I_2 = (n \log n)m^{n-1} I' + mn$ and $E_2 = (n \log n)m^{n-1}E' + mn$, and thus $O(n \log n)$ of Algorithm 1 can be ignored. Therefore, CRM dramatically reduces the time complexity (i.e., the term $2^n$ in $I_1$ and $E_1$ is changed to the term $n \log n$ in $I_2$ and $E_2$).

## 4 Experiments

Following prior work for NEs [34, 4, 13], we evaluate our approach on two sets of games: randomly generated games (i.e., $(n, m)$ with $n$ players and $m$ actions for each player) and six-player three-action games that are generated by GAMUT [31]. Payoffs are generated from the interval between 0 and 100 (other ranges (e.g., $[0, 1]$) do not affect the result). Details are shown in Appendix F. We show the game size in terms of the number of bilinear terms and integer variables in Appendix G, e.g., the number of bilinear terms in the game $(9, 2)$ is 19152 based on $\overline{\mathcal{N}}$ but is 2512 based on $\underline{\mathcal{N}}$.

**Baselines:** We compare our CRM[4] shown in Algorithm 2, i.e., solving Program (13) based on our $\underline{\mathcal{N}}$, to the state-of-the-art algorithms: i) **MIBP** [34, 13]: the equivalent of solving Program (5) based on $\overline{\mathcal{N}}$; ii) **EXCLUSION** [4]: the first implemented algorithm guarantees to converge to an NE by using a tree-search based method by splitting the continuous probability space of the solution; and iii) **ENUMPOLY** [28]: an algorithm in the well-known game-solving package Gambit which

---

[4]Codes are available at https://github.com/Youzhi333/optimalNE.

tries to find all NEs by enumerating all the supports which could be the support of an NE and then searching for an equilibrium on that support. They represent approaches to solving a nonlinear program, finding an NE, and enumerating all Nash equilibria, respectively. There are some other algorithms in Gambit [28] for finding an NE in a multiplayer game, including: i) **GNM** [16]: a global Newton method approach; ii) **IPA** [17]: an iterated polymatrix approximation approach; iii) **LIAP**: a function minimization approach; iv) **SIMPDIV** [41]: a simplicial subdivision approach; and v) **LOGIT** [29, 40]: a quantal response method. However, they cannot guarantee finding an NE [4]. Therefore, they are not suitable for finding an optimal NE. In fact, we show in Appendix I that all of them fail to solve many games and even run significantly slower than CRM in many games. Note that these Gambit algorithms only achieve some NE if the game is solved, which may not be optimal.

**Algorithm Setting and Metric:** We set a time limit of 1000 seconds for each case unless stated otherwise. Our optimality gap for EXCLUSION is significantly smaller than 0.001 in [4] (we verified that, with the same optimality gap, our result for EXCLUSION is almost the same as the one in [4]). We mainly use the runtime and the percentage of games that are not solved within the time limit to measure the performance of our approach. Details are shown in Appendix F (also caption of Table 1).

**Result:** Part of results are shown in Table 1, and more results are in Tables 4 and 5 of Appendix H. They show that the runtime of our CRM steadily increases with the game size. Note that the runtime of CRM includes the runtime for our Algorithm 1, which is extremely small (see Appendix E). Moreover, CRM is significantly faster than the baselines and is two or three orders of magnitude faster than the state-of-art baselines MIBP, ENUMPOLY, and EXCLUSION in most games. The reasons are that: 1) MIBP with too many bilinear terms and large feasible solution space after the relaxation cannot perform well without CRM's novel techniques in Section 3, where each of these techniques significantly boosts the performance (see the ablation study); and 2) the exponentially many NEs and the large search space caused by splitting the continuous probability space make ENUMPOLY and EXCLUSION, respectively, hard to scale up. EXCLUSION always has large utility gaps, which means that we will lose large utilities if we use EXCLUSION for our problem. The result that CRM runs significantly faster than EXCLUSION means that CRM is a faster algorithm not only for computing an optimal NE but also for just computing an NE. Furthermore, the gap between CRM and any of the baselines increases with the number of players or actions. In games with a large gap between CRM and baselines, the real gap should be larger because these baselines have not solved all of them within the time limit, while CRM solved all of them. Overall, CRM significantly overcomes the limitation of baselines.

**Ablation Study**: We evaluate each component of CRM by using the following variants: i) **CR**: solving Program (13) based on $\overline{\mathcal{N}}$; ii) **CM**: solving Program (13) based on $\underline{\mathcal{N}}$ without the relation constraints Eqs.(11) and (12); iii) **C**: solving Program (13) based on $\overline{\mathcal{N}}$ without the relation constraints Eqs.(11) and (12); and iv) **M**: solving Program (5) based on $\underline{\mathcal{N}}$. Part of results are in Table 2, and more results are in Table 6 of Appendix H. We can see that each component of our approach significantly boosts its performance.

## 5 Related Work

Existing works define a correlation plan as a probability distribution over the joint action space of all players, and use it to formulate constraints for a correlated equilibrium [37, 1]. However, the constraints for the space of correlated equilibria cannot be used in our program due to the following two reasons. First, there are no correlation plans for coordinating all players in our program after the convex relaxation because our formulation based on [34] has reduced the degree of the multilinear program for the space of NEs in order to significantly reduce the number of bilinear terms. Second, our correlation plans are different from the correlation plan for correlated equilibria because our correlation plans are only for subsets of the players. Recently, the correlation plan [49] based on a decomposition of the extensive-form game into public states has been used to compute correlated equilibria. However, their approach is not suitable for our problem because our game is not extensive-form and then does not have the property of their problem. Then our approach exploiting the relations of correlation plans and minimizing the number of correlation plans is novel.

Several recent efforts have developed relatively efficient algorithms to find an NE that maximizes the utility of a team of players in zero-sum games [45, 50, 51, 52, 11, 24, 53]. However, these algorithms cannot be used in games where team members have different utility functions. Existing

works transforming multilinear terms into bilinear terms only focus on special cases. For example, the transformation in [13] is equivalent to our transformation based on $\overline{\mathcal{N}}$, which is only a special case of our transformation framework. They [13] then directly solves the bilinear program based on this special transformation for finding an NE, which is equivalent to our baseline MIBP. Experiments show that our approach with novel techniques in Section 3 significantly outperforms [13]. Similar to the formulation in [34], there are other formulations [2, 3] for finding an optimal NE for two players under the problem of computing a leader-follower (Stackelberg) equilibrium for a single leader and two followers after a mixed strategy is committed by the leader. These formulations are different from ours because of the difference between the NE and the Stackelberg equilibrium. For example, the leader will commit a strategy to the followers in a Stackelberg equilibrium, i.e., the followers know the leader's strategy, but this cannot happen in an NE as all players move simultaneously. Moreover, after dropping the dependences of the followers to the leader's strategies in these bilinear programming formulations, the problem boils down to computing an optimal NE in two-player games because they only consider two followers in their formulations, which results in the same two-player formulation of [34].

For the existing general optimization techniques, e.g., Reformulation-Linearization Technique (RLT) [35, 25, 36], they add linear constraints by multiplying linear constraints with a single variable to reduce the feasible solution space of the convex relaxation and the number of bilinear terms if they can be represented by linear constraints (i.e., variants of original linear constraints). However, these operations are not very effective for our problem because the bilinear terms cannot be represented by those linear constraints (i.e., variants of original linear constraints), and simply multiplying linear constraints with a single variable cannot effectively represent the relation between auxiliary variables and nonlinear terms. Indeed, RLT is implemented in Gurobi [18, 19], but its performance (see MIBP in Table 1) is not good enough for large games in experiments. Moreover, our approach significantly outperforms the state-of-the-art optimization solver Gurobi (see results for CRM versus MIBP in Table 1) in experiments.

# 6 Limitations

Similarly to the previous literature [34, 4, 13], to efficiently evaluate the algorithms, we set a time limit of 1000 seconds for each case unless stated otherwise. It means that we may need 30,000 seconds (almost 8 hours) to run an algorithm for each game setting (e.g., the game $(6,3)$) with 30 cases. We totally run 13 algorithms for 21 different game settings, whose total runtime is more than 2,000 hours if each algorithm needs 30,000 seconds for each game setting. A higher time limit means more runtime. For example, if the time limit is 10,000s, we may need $20,000$ hours (more than 800 days), which is not reasonable for a personal computer. Increasing the game size will cause a similar problem as well. Our goal is only to show that our proposed algorithm runs faster than baselines. Therefore, as a proof of concept, our time limit and game size are reasonable and practical.

Our algorithm CRM is significantly faster than the state-of-the-art baseline, and it can solve many real-world games: e.g., (1) multiplayer hand games using only the hands of the players (https://en.m.wikipedia.org/wiki/Hand_game), including the rock-paper-scissors games, Morra games, and their variants; and (2) the matching pennies game with several players and only two actions for each player. However, we cannot handle extremely large games now because we are handling a very hard problem, and then it is unrealistic to expect that our exact algorithm CRM could run very fast in large games. Our algorithm is an attempt to make this computation of optimal NEs feasible, and our algorithm framework can be built on by further innovative heuristics to improve the computation of optimal NEs. That is, for games with more players or actions, we can exploit the auxiliary speed-up techniques: the multiagent learning framework—Policy-Spaced Response Oracles (PSRO) [22, 54, 55, 51, 46, 23], the abstraction techniques [47], or only considering approximate NEs. Specifically, our algorithm CRM could be used as the meta-solver in PSRO.

# 7 Conclusion

This paper proposes a novel algorithm (CRM) for computing optimal NEs based on our transformation framework. CRM uses correlation plans with their relations to strictly reduce the feasible solution space after the convex relaxation while minimizing the number of correlation plans to significantly reduce the number of bilinear terms. Experiments show that CRM significantly outperforms baselines.

## Acknowledgments and Disclosure of Funding

This research is supported by the InnoHK Fund, ONR grants N00014-18-1-2670, and N00014-20-1-2407. Bo An is supported by the National Research Foundation, Singapore under its Industry Alignment Fund – Pre-positioning (IAF-PP) Funding Initiative. Any opinions, findings and conclusions or recommendations expressed in this material are those of the author(s) and do not reflect the views of National Research Foundation, Singapore.

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

# Appendix

## A  Illustration of CRM

In four-player games,

$$\overline{\mathcal{N}} = \{\{1,2,3\},\{1,2,4\},\{1,3,4\},\{2,3,4\},\{1,2\},\{1,3\},\{2,3\},\{1,4\},\{3,4\},\{2,4\}\}.$$

By Algorithm 1 in CRM, we have:

$$\underline{\mathcal{N}} = \{\{1,2,3\},\{4,2,3\},\{1,4,3\},\{1,2,4\},\{1,2\},\{4,2\},\{1,4\}\},$$

i.e., $\underline{\mathcal{N}} = \{-4,-1,-2,-3,\{1,2\},\{2,4\},\{1,4\}\}$. To obtain this set, we first have a full binary tree $T_{-4}$ with the set of internal nodes $\mathcal{N}_{T_{-4}} = \{\{1,2,3\},\{1,2\}\}$. Then, for each element in $\mathcal{N}_{T_{-4}}$, we replace 1 with 4 to obtain $\mathcal{N}_{T_{-1}} = \{\{4,2,3\},\{4,2\}\}$; replace 2 with 4 to obtain $\mathcal{N}_{T_{-2}} = \{\{1,4,3\},\{1,4\}\}$; and replace 3 with 4 to obtain $\mathcal{N}_{T_{-3}} = \{\{1,2,4\},\{1,2\}\}$. Then $\underline{\mathcal{N}} = \mathcal{N}_{T_{-1}} \cup \mathcal{N}_{T_{-2}} \cup \mathcal{N}_{T_{-3}} \cup \mathcal{N}_{T_{-4}}$.

In three-player games, Algorithm 1 cannot reduce the number of internal nodes because each element in $\{-i \mid i \in \{1,2,3\}\}$ includes only two players. Then $\overline{\mathcal{N}} = \underline{\mathcal{N}} = \{\{1,2\},\{1,3\},\{2,3\}\}$, which means that $\overline{\mathcal{N}}$ and $\underline{\mathcal{N}}$ will result in the same set of bilinear terms in three-player games.

To be simplified, we show how to formulate the program according to Algorithm 2 (i.e., CRM) in a three-player game $G = (N, A, u)$ with $N = \{1,2,3\}$, $A_i = \{a_i, a_i'\}$ first. In three-player games, $-1 = \{2,3\}$, and $\underline{\mathcal{N}} = \{-1,-2,-3\} = \{\{2,3\},\{1,3\},\{1,2\}\}$.

Note that $A_{-1} = A_{2,3} = A_2 \times A_3 = \{(a_2,a_3),(a_2,a_3'),(a_2',a_3),(a_2',a_3')\}$ with $N' = -1 = \{2,3\}$, $N_l' = \{2\}$, $N_r' = \{3\}$. We show the constraints related to player 1 here: We first show player 1's constraints in the Nash equilibria space based on Eqs.(1b)-(1e):

$x_1(a_1) + x_1(a_1') = 1$ **(based on Eqs.(1b))**

$1 - b_{a_1} \geq x_1(a_1), 1 - b_{a_1'} \geq x_1(a_1')$ **(based on Eq.(1c))**

$u_1(x) \geq u_1(a_1, x_{-1}), u_1(x) - u_1(a_1, x_{-1}) \leq b_{a_1}(U_{max} - U_{min})$ **(based on Eqs.(1d)) and (1e))**

$u_1(x) \geq u_1(a_1', x_{-1}), u_1(x) - u_1(a_1', x_{-1}) \leq b_{a_1'}(U_{max} - U_{min})$ **(based on Eqs.(1d) and (1e))**

The above constraints include player 1's expected utility variables $u_1(a_1, x_{-1})$ and $u_1(a_1', x_{-1})$, which are represented by the following constraints based on Eq.(4):

$$u_1(a_1, x_{-1}) = u_1(a_1, a_2, a_3)x_{-1}(a_2, a_3) + u_1(a_1, a_2, a_3')x_{-1}(a_2, a_3')$$
$$+ u_1(a_1, a_2', a_3)x_{-1}(a_2', a_3) + u_1(a_1, a_2', a_3')x_{-1}(a_2', a_3') \text{ (based on Eq.(4))}$$
$$u_1(a_1', x_{-1}) = u_1(a_1', a_2, a_3)x_{-1}(a_2, a_3) + u_1(a_1', a_2, a_3')x_{-1}(a_2, a_3')$$
$$+ u_1(a_1', a_2', a_3)x_{-1}(a_2', a_3) + u_1(a_1', a_2', a_3')x_{-1}(a_2', a_3') \text{ (based on Eq.(4))} ,$$

where the correlation plan $x_{-1}$ of $-1$ over $A_{-1}$, based on Eq.(2), is defined by:

$$x_{-1}(a_2, a_3) + x_{-1}(a_2, a_3') + x_{-1}(a_2', a_3) + x_{-1}(a_2, a_3') = 1 \text{ (the plan of } -1 \text{ based on Eq.(2))}.$$

In the above correlation plan $x_{-1}$, $x_{-1}(a_2, a_3)$, $x_{-1}(a_2, a_3')$, $x_{-1}(a_2', a_3)$, $x_{-1}(a_2', a_3')$ represent the following four bilinear terms (constraints):

$x_{-1}(a_2, a_3) = x_2(a_2)x_3(a_3)$ **(the bilinear constraint based on Eq.(3a)) with**
$$a_{-1} = (a_2, a_3) = (a_{N_l'}, a_{N_r'}), a_{N_l'} = (a_2), a_{N_r'} = (a_3)$$

$x_{-1}(a_2, a_3') = x_2(a_2)x_3(a_3')$ **(the bilinear constraint based on Eq.(3a)) with**
$$a_{-1} = (a_2, a_3') = (a_{N_l'}, a_{N_r'}), a_{N_l'} = (a_2), a_{N_r'} = (a_3')$$

$x_{-1}(a_2', a_3) = x_2(a_2')x_3(a_3)$ **(the bilinear constraint based on Eq.(3a)) with**
$$a_{-1} = (a_2', a_3) = (a_{N_l'}, a_{N_r'}), a_{N_l'} = (a_2'), a_{N_r'} = (a_3)$$

$x_{-1}(a_2', a_3') = x_2(a_2')x_3(a_3')$ **(the bilinear constraint based on Eq.(3a)) with**
$$a_{-1} = (a_2', a_3') = (a_{N_l'}, a_{N_r'}), a_{N_l'} = (a_2'), a_{N_r'} = (a_3'),$$

where, based on the binary division in $\underline{\mathcal{N}}$, any joint action $a_{N'} \in A_{N'}$ can be divided into two sub-joint actions $a_{N'_l} \in A_{N'_l}$ and $a_{N'_r} \in A_{N'_r}$ such that $a_{N'} = (a_{N'_l}, a_{N'_r})$. Note that $x_2$ and $x_3$ are special correlation plans with that 2 is set to be equivalent to $\{2\}$ and 3 is set to be equivalent to $\{3\}$ for simplification. Correlation plans $x_{-1}$ and $x_2$ (or $x_3$) have the following relation:

$$x_{-1}(a_2, a_3) + x_{-1}(a_2, a'_3) = x_2(a_2) \text{ (the relation of correlation plans } x_{-1} \text{ and } x_2 \text{ based on Eq.(11)),}$$

$$\forall a_{N'} \in \{(a_2, a_3), (a_2, a'_3)\} \subseteq A_{-1}, a_{N'}(2) = a_2, \textbf{ i.e., } a_2 \textbf{ is player 2's action in } a_{N'}$$

$$x_{-1}(a'_2, a_3) + x_{-1}(a'_2, a'_3) = x_2(a'_2) \text{ (the relation of correlation plans } x_{-1} \text{ and } x_2 \text{ based on Eq.(11)),}$$

$$\forall a_{N'} \in \{(a'_2, a_3), (a'_2, a'_3)\} \subseteq A_{-1}, a_{N'}(2) = a'_2, \textbf{ i.e., } a'_2 \textbf{ is player 2's action in } a_{N'}$$

$$x_{-1}(a_2, a_3) + x_{-1}(a'_2, a_3) = x_3(a_3) \text{ (the relation of correlation plans } x_{-1} \text{ and } x_3 \text{ based on Eq.(11)),}$$

$$\forall a_{N'} \in \{(a_2, a_3), (a'_2, a_3)\} \subseteq A_{-1}, a_{N'}(3) = a_3, \textbf{ i.e., } a_3 \textbf{ is player 3's action in } a_{N'}$$

$$x_{-1}(a_2, a'_3) + x_{-1}(a'_2, a'_3) = x_3(a'_3) \text{ (the relation of correlation plans } x_{-1} \text{ and } x_3 \text{ based on Eq.(11)),}$$

$$\forall a_{N'} \in \{(a_2, a'_3), (a'_2, a'_3)\} \subseteq A_{-1}, a_{N'}(3) = a'_3, \textbf{ i.e., } a'_3 \textbf{ is player 3's action in } a_{N'}$$

Constraints related to other players are created similarly. As shown in Algorithm 2 (i.e., CRM), after creating all of these constraints in the bilinear program for the Nash equilibria space, we can solve the program to optimize an objective function by using a global optimization solver, e.g., Gurobi.

In three-player games, $x_{-1}(a_{-1})$ just represents a bilinear term, and then a chain of bilinear constraints (equalities) to transform a multilinear term into bilinear terms is not explicit. In four-player games, a chain of bilinear constraints (equalities) to transform a multilinear term into bilinear terms is more explicit. For example, in a game $G = (N, A, u)$ with $N = \{1, 2, 3, 4\}$, $A_i = \{a_i, a'_i\}$, and $\underline{\mathcal{N}} = \{-4, -1, -2, -3, \{1, 2\}, \{2, 4\}, \{1, 4\}\}$, based on Eq.(3a), we have:

$$x_{-1}(a_2, a_3, a_4) = x_{2,4}(a_2, a_4)x_3(a_3), x_{2,4}(a_2, a_4) = x_2(a_2)x_4(a_4) \text{ (a chain of bilinear constraints),}$$

where $N' = -1 = \{2, 3, 4\}$, $N'_l = \{2, 4\}$ and $N'_r = \{3\}$ based on $\underline{\mathcal{N}}$. That is, $a_{-1} = (a_2, a_3, a_4) = (a_{N'_l}, a_{N'_r})$ with $a_{N'_l} = (a_2, a_4), a_{N'_r} = (a_3)$ (i.e., joint action $a_{-1}$ is divided into two sub-joint actions $a_{N'_l}$ and $a_{N'_r}$ such that $a_{-1} = (a_{N'_l}, a_{N'_r})$) based on $\underline{\mathcal{N}}$, and then we can have the above chain of bilinear constraints for it. Each joint action in $A_{-1} = A_2 \times A_3 \times A_4 = A_{N'_l} \times A_{N'_r} = \{(a_2, a_3, a_4), (a_2, a_3, a'_4), (a_2, a'_3, a_4), (a_2, a'_3, a'_4), (a'_2, a_3, a_4), (a'_2, a_3, a'_4), (a'_2, a'_3, a_4), (a'_2, a'_3, a'_4)\}$ is divided into two disjoint sets similarly, and then we can have a chain of bilinear constraints for it.

In three-player games, each element in $\{-i \mid i \in \{1, 2, 3\}\}$ includes only two players, and then the resulting program does not include the constraints for the relation of correlation plans based on Eq.(12). To show the constraints based on Eq.(12), we consider four-player games. The following constraints are player 1's constraints based on Eq.(12) for solving a game $G = (N, A, u)$ with $N = \{1, 2, 3, 4\}$, $A_i = \{a_i, a'_i\}$:

$$x_{-1}(a_2, a_3, a_4) + x_{-1}(a_2, a'_3, a_4) = x_{2,4}(a_2, a_4) \textbf{ (based on Eq.(12)) with } a_{N'_l} = (a_2, a_4),$$

$$\forall a_{N'} \in \{(a_2, a_3, a_4), (a_2, a'_3, a_4)\} \subseteq A_{-1}, a_{N'}(N'_l) = (a_2, a_4),$$

$$\textbf{i.e., } (a_2, a_4) \textbf{ is } N'_l\textbf{'s sub-joint action in } a_{N'}$$

$$x_{-1}(a_2, a_3, a'_4) + x_{-1}(a_2, a'_3, a'_4) = x_{2,4}(a_2, a'_4) \textbf{ (based on Eq.(12)) with } a_{N'_l} = (a_2, a'_4),$$

$$\forall a_{N'} \in \{(a_2, a_3, a'_4), (a_2, a'_3, a'_4)\} \subseteq A_{-1}, a_{N'}(N'_l) = (a_2, a'_4),$$

$$\textbf{i.e., } (a_2, a_4) \textbf{ is } N'_l\textbf{'s sub-joint action in } a_{N'}$$

$$x_{-1}(a'_2, a_3, a_4) + x_{-1}(a'_2, a'_3, a_4) = x_{2,4}(a'_2, a_4) \textbf{ (based on Eq.(12)) with } a_{N'_l} = (a'_2, a_4),$$

$$\forall a_{N'} \in \{(a'_2, a_3, a_4), (a'_2, a'_3, a_4)\} \subseteq A_{-1}, a_{N'}(N'_l) = (a'_2, a_4),$$

$$\textbf{i.e., } (a_2, a_4) \textbf{ is } N'_l\textbf{'s sub-joint action in } a_{N'}$$

$$x_{-1}(a'_2, a_3, a'_4) + x_{-1}(a'_2, a'_3, a'_4) = x_{2,4}(a'_2, a'_4) \textbf{ (based on Eq.(12)) with } a_{N'_l} = (a'_2, a'_4),$$

$$\forall a_{N'} \in \{(a'_2, a_3, a'_4), (a'_2, a'_3, a'_4)\} \subseteq A_{-1}, a_{N'}(N'_l) = (a'_2, a'_4),$$

$$\textbf{i.e., } (a'_2, a'_4) \textbf{ is } N'_l\textbf{'s sub-joint action in } a_{N'},$$

where $N' = -1 = \{2, 3, 4\}$, $N'_l = \{2, 4\}$ and $N'_r = \{3\}$ based on $\underline{\mathcal{N}}$. The we can divide the joint actions in $A_{-1}$, e.g., $a_{-1} = (a_2, a_3, a_4) = (a_{N'_l}, a_{N'_r})$ with $a_{N'_l} = (a_2, a_4), a_{N'_r} = (a_3)$. For elements in $\{N'_l, N'_r\}$, we only consider constraints based on Eq.(12) for $N'_l$ because $|N'_l| = 2 > 1$

and $|N'_r| = 1$. We could use Eq.(11) to generate constraints for $N'_r = \{3\}$ for the relation between $x_{-1}$ and $x_3$, e.g., for $a_3 \in A_3$,

$$x_{-1}(a_2, a_3, a_4) + x_{-1}(a'_2, a_3, a_4) + x_{-1}(a_2, a_3, a'_4) + x_{-1}(a'_2, a_3, a'_4) = x_3(a_3),$$

where, for each $a_{N'} \in \{(a_2, a_3, a_4), (a'_2, a_3, a_4), (a_2, a_3, a'_4), (a'_2, a_3, a'_4)\} \subseteq A_{-1}, a_{N'}(3) = a_3$, i.e., $a_3$ is player 3's action in $a_{N'}$.

Other constraints for four-player games are created similarly to the above creation of constraints for three-player games.

## B  Proofs

**Theorem 1.** *The feasible solution space of mixed strategies (i.e., $x_i(a_i)$ for each $i \in N$, $a_i \in A_i$) in Eqs.(1b)-(1g), (3), and (4) is the space of NEs.*

*Proof.* Eq.(1) describing the space of NEs and Eqs.(1b)-(1g), (3), and (4) both include Eqs.(1b)-(1g), which describe the condition of NEs. Then we only need to show that Eq.(4) is equivalent to Eq.(1a). That is, we need to show $\prod_{j\in-i} x_j(a_{-i}(j)) = x_{-i}(a_{-i})$ for each $i \in N$ and $a_{-i} \in A_{-i}$. To achieve this result, we show $\prod_{j\in N'} x_j(a_{N'}(j)) = x_{N'}(a_{N'})$ for each $N' \in \mathcal{N}$ and $a_{N'} \in A_{N'}$. We show that this statement holds by induction. For any $N' \in \mathcal{N}$ with $|N'| = 2$, we obviously have $\prod_{j\in N'} x_j(a_{N'}(j)) = x_{N'}(a_{N'})$ for each $a_{N'} \in A_{N'}$ by Eq.(3). Suppose, for any $N' \in \mathcal{N}$ with $|N'| = k \geq 2$, the statement holds. Now for any $N' \in \mathcal{N}$ with $|N'| = k + 1$ and its two children $N'_l$ and $N'_r$, we have: for any $a_{N'} \in A_{N'}$,

$$x_{N'}(a_{N'}) = x_{N'_l}(a_{N'_l}) x_{N'_r}(a_{N'_r})$$
$$= \prod_{i\in N'_l} x_i(a_{N'_l}(i)) \prod_{j\in N'_r} x_j(a_{N'_r}(j))$$
$$= \prod_{j\in N'} x_j(a_{N'}(j)),$$

where the second "=" is based on the assumption that, for any $N' \in \mathcal{N}$ with $|N'| = k \geq 2$, the statement holds. Then $\prod_{j\in N'} x_j(a_{N'}(j)) = x_{N'}(a_{N'})$ for each $N' \in \mathcal{N}$ and $a_{N'} \in A_{N'}$. Therefore, the theorem holds. □

**Theorem 2.** $\mathcal{M} \subset \mathcal{T} \subset \mathcal{R}$, *i.e., $\mathcal{T}$ is strictly smaller than $\mathcal{R}$ but still includes $\mathcal{M}$.*

*Proof.* (i) We first show that $\mathcal{T} \subset \mathcal{R}$. Given any $x_{\{i,j\}}(a_i, a_j) = x_i(a_i)x_j(a_j)$, in $\mathcal{T}$, we have $x_{\{i,j\}}(a_i, a_j) \leq \min\{x_i(a_i), x_j(a_j)\}$ according to Eq.(11). Suppose $x_i(a_i) + x_j(a_j) - 1 > x_{\{i,j\}}(a_i, a_j)$. According to Eqs.(11) and (1b), we have the following contradiction:

$$x_i(a_i)$$
$$= \sum_{a'_j \in A_j} x_{\{i,j\}}(a_i, a'_j)$$
$$< x_i(a_i) + x_j(a_j) - 1 + \sum_{a'_j \in A_j, a'_j \neq a_j} x_{\{i,j\}}(a_i, a'_j)$$
$$\leq x_i(a_i) - 1 + \sum_{a_j \in A_j} x_j(a_j)$$
$$= x_i(a_i),$$

where the first "=" is according to Eq.(11), "<" is according to the assumption, "≤" is according to Eq.(11), and the last "=" is according to Eq.(1b). This contradiction implies that $x_i(a_i) + x_j(a_j) - 1 \leq x_{\{i,j\}}(a_i, a_j)$, i.e., $\max\{0, x_i(a_i) + x_j(a_j) - 1\} \leq x_{\{i,j\}}(a_i, a_j)$. Similarly, for any $x_{N'}(a_{N'}) = x_{N'_l}(a_{N'_l}) x_{N'_r}(a_{N'_r})$ with two children $N'_l$ and $N'_r$ of $N'$, in $\mathcal{T}$, we have:

$$\max\{x_{N'_l}(a_{N'_l}) + x_{N'_r}(a_{N'_r}) - 1, 0\}$$
$$\leq x_{N'}(a_{N'})$$
$$\leq \min\{x_{N'_l}(a_{N'_l}), x_{N'_r}(a_{N'_r})\}.$$

Therefore, $\mathcal{T} \subseteq \mathcal{R}$.

Given any $x_{\{i,j\}}(a_1, a_2) = x_1(a_1)x_2(a_2)$ and $x_{\{i,j\}}(a_1', a_2) = x_1(a_1')x_2(a_2)$, by Eq.(6), we can have a feasible solution such that:

$$x_{\{i,j\}}(a_1, a_2) = \min\{x_1(a_1), x_2(a_2)\}$$
$$x_{\{i,j\}}(a_1', a_2) = \min\{x_1(a_1'), x_2(a_2)\}.$$

Then $x_{\{i,j\}}(a_1, a_2) + x_{\{i,j\}}(a_1', a_2) > x_2(a_2)$ when $0 < x_2(a_2) < \min\{x_1(a_1), x_1(a_1')\} < 1$. However, in $\mathcal{T}$,

$$x_{\{i,j\}}(a_1, a_2) + x_{\{i,j\}}(a_1', a_2) \leq x_2(a_2).$$

Then $\mathcal{R} \nsubseteq \mathcal{T}$. Therefore, $\mathcal{T} \subset \mathcal{R}$, i.e., $\mathcal{T}$ is strictly smaller than $\mathcal{R}$.

(ii) Now we show that $\mathcal{M} \subset \mathcal{T}$. In $\mathcal{M}$, for each $a_{N'} \in A_{N'}, N' \in \mathcal{N}$ with two children $N_l'$ and $N_r'$ of $N'$, there is a bilinear constraint $x_{N'}(a_{N'}) = x_{N_l'}(a_{N_l'})x_{N_r'}(a_{N_r'})$ based on $a_{N'} = (a_{N_l'}, a_{N_r'})$, where $x_{N'}(a_{N'}) = x_i(a_i)$ for $N' = \{i\}$. We first show $\mathcal{M} \subseteq \mathcal{T}$ for Eq.(2) by induction. For any $N' = (i, j) \in \mathcal{N}$, by $\mathcal{M}$, we have:

$$\sum_{(a_i, a_j) \in A_{\{i,j\}}} x_{\{i,j\}}(a_i, a_j) = \sum_{a_i \in A_i} x_i(a_i) \sum_{a_j \in A_j} x_j(a_j) = 1.$$

Suppose, for any $N' \in \mathcal{N}$ with $|N'| = k$ and $k \geq 2$,

$$\sum_{a_{N'} \in A_{N'}} x_{N'}(a_{N'}) = 1.$$

Now for any $N' \in \mathcal{N}$ with $|N'| = k + 1$, with two children $N_l'$ and $N_r'$ of $N'$, by $\mathcal{M}$ and the assumption of $N_k$, we have:

$$\sum_{a_{N'} \in A_{N'}} x_{N'}(a_{N'}) = \sum_{a_{N_l'} \in A_{N_l'}} x_{N_l'}(a_{N_l'}) \sum_{a_{N_r'} \in A_{N_r'}} x_{N_r'}(a_{N_r'})$$
$$= 1.$$

Therefore, Eq.(2) is implied by $\mathcal{M}$. Similarly, for each $N' \subset N$, we have $\sum_{a_{N'} \in A_{N'}} x_{N'}(a_{N'}) = 1$.

For any $N' = (i, j) \in \mathcal{N}$, by $\mathcal{M}$, we have: for any $a_i \in A_i$,

$$\sum_{a_j \in A_j} x_{\{i,j\}}(a_i, a_j) = x_i(a_i) \sum_{a_j \in A_j} x_j(a_j) = x_i(a_i);$$

and for any $a_j \in A_j$, $\sum_{a_i \in A_i} x_{\{i,j\}}(a_i, a_j) = x_j(a_j) \sum_{a_i \in A_i} x_i(a_i) = x_j(a_j)$. For any $N' \in \mathcal{N}$ with $|N'| > 2$, by $\mathcal{M}$, we have: for any $i \in N'$, and $a_i \in A_i$, with $N_k = N' \setminus \{i\}$,

$$\sum_{a_{N'} \in A_{N'}, a_{N'}(i) = a_i} x_{N'}(a_{N'}) = x_i(a_i) \sum_{a_{N_k} \in A_{N_k}} x_{N_k}(a_{N_k})$$
$$= x_i(a_i).$$

Therefore, Eq.(11) is implied by $\mathcal{M}$.

Now for any $N' \in \mathcal{N}$, with two children $N_l'$ and $N_r'$ of $N'$, by $\mathcal{M}$, for each $a_{N_l'} \in A_{N_l'}$, we have:

$$\sum_{a_{N'} = (a_{N_l'}, a_{N_r'}) \in A_{N'}} x_{N'}(a_{N'})$$
$$= x_{N_l'}(a_{N_l'}) \sum_{a_{N_r'} \in A_{N_r'}} x_{N_r'}(a_{N_r'})$$
$$= x_{N_l'}(a_{N_l'}),$$

where the condition $a_{N'} = (a_{N'_l}, a_{N'_r}) \in A_{N'}$ represents that $a_{N'} \in A_{N'}, a_{N'}(N'_l) = a_{N'_l}$. For each $a_{N'_r} \in A_{N'_r}$, we have:

$$\sum_{a_{N'} = (a_{N'_l}, a_{N'_r}) \in A_{N'}} x_{N'}(a_{N'})$$

$$= x_{N'_r}(a_{N'_r}) \sum_{a_{N'_l} \in A_{N'_l}} x_{N'_l}(a_{N'_l})$$

$$= x_{N'_r}(a_{N'_r}).$$

Therefore, Eq.(12) is implied by $\mathcal{M}$. Then $\mathcal{M} \subseteq \mathcal{T}$.

Given any four bilinear terms:

$$x_{\{1,2\}}(a_1, a_2) = x_1(a_1)x_2(a_2)$$
$$x_{\{1,2\}}(a'_1, a_2) = x_1(a'_1)x_2(a_2)$$
$$x_{\{1,2\}}(a_1, a'_2) = x_1(a_1)x_2(a'_2)$$
$$x_{\{1,2\}}(a'_1, a'_2) = x_1(a'_1)x_2(a'_2).$$

The following solution is in $\mathcal{T}$:

$$x_{\{1,2\}}(a_1, a_2) = 0$$
$$x_{\{1,2\}}(a'_1, a_2) = 2/3$$
$$x_{\{1,2\}}(a_1, a'_2) = 1/3$$
$$x_{\{1,2\}}(a'_1, a'_2) = 0$$
$$x_1(a_1) = 1/3, x_1(a'_1) = 2/3$$
$$x_2(a_2) = 2/3, x_2(a'_2) = 1/3.$$

However, the above solution is not in $\mathcal{M}$ because: $x_1(a_1) = 1/3$ and $x_2(a_2) = 2/3$ imply that $x_{\{1,2\}}(a_1, a_2) = 2/9$, which contradicts $x_{\{1,2\}}(a_1, a_2) = 0$ in the above solution. Therefore, $\mathcal{T} \nsubseteq \mathcal{M}$. That is, $\mathcal{M} \subset \mathcal{T}$. □

**Theorem 3.** *The optimal solution of Program (13) maximizes $g(x)$ over the space of NEs.*

*Proof.* By Theorem 2, $\mathcal{T}$ includes $\mathcal{M}$, i.e., $\mathcal{T}$ does not reduce the space of NEs. Program (13) is obtained after we explicitly restrict the feasible solution space to $\mathcal{T}$ by adding Eqs.(2), (11), and (12) to Program (5). The optimization solver will search this feasible solution space after the relaxation to find the optimal solution for the original bilinear program. Therefore, by solving Program (13), we obtain an optimal NE. □

**Theorem 4.** $\underline{\mathcal{N}}$ *generated by Algorithm 1 is a binary collection, and $O(n \log n)$ for the size of $\underline{\mathcal{N}}$ is the minimum size of all binary collections of a game $G$.*

*Proof.* First, it is clear that $\underline{\mathcal{N}}$ generated by Algorithm 1 is a binary collection of $G$. Then $\{-i \mid i \in N\} \subseteq \underline{\mathcal{N}}$.

The number of internal nodes in each binary tree with $n-1$ leaves of $-i$ for each $i \in N$ is $n-2$ [20]. To obtain the minimum number of internal nodes in these binary trees for $\{-i \mid i \in N\}$, we can minimize the difference between binary trees. Given a binary tree $T_{-n}$ for $-n$ and a binary tree $T_{-i}$ for $-i$ with $i \in -n$, the difference between $T_{-n}$ and $T_{-i}$ at least includes the path from the root to the node $\{i\}$ in $T_{-n}$ and the path from the root to the node $\{n\}$ in $T_{-i}$. Then the number of different internal nodes (i.e., nodes that are not in $T_{-n}$) in these binary trees for $\{-i \mid i \in N\}$ is at least equal to the total path length in $T_{-n}$. Algorithm 1 ensures that the number of different internal nodes in these binary trees for $\{-i \mid i \in N\}$ is equal to the total path length in $T_{-n}$, and the total path length in $T_{-n}$ by Algorithm 1 is at most $(n-1)\lceil \log_2(n-1) \rceil$. Given a binary tree with $k-1$ internal nodes, the minimum total path length is $O(k \log k)$ [20]. Therefore, $O(n \log n)$ for the size of $\underline{\mathcal{N}}$ is the minimum size of all binary collections of $G$. □

**Algorithm 3** Generate $\underline{\mathcal{N}}$: full details of Algorithm 1

---

1: $Build(-n)$
2: **for** each $i$ in $\{1,\ldots,n-1\}$ **do**
3:    $-i$: replace $i$ with $n$ in $-n$
4:    $N' \leftarrow -n$
5:    $\underline{\mathcal{N}} \leftarrow \underline{\mathcal{N}} \cup \{-i\}$
6:    **while** $|N'| > 2$ **do**
7:       $\{N_1, N_2\} \leftarrow Ch(N')$ with $i \in N_1$
8:       $N''$: replace $i$ with $n$ in $N'$ ;
9:       $N_1'$: replace $i$ with $n$ in $N_1$
10:      $Ch(N'') \leftarrow \{N_1', N_2\}$
11:      **if** $|N_1| > 1$ **then**
12:         $\underline{\mathcal{N}} \leftarrow \underline{\mathcal{N}} \cup \{N_1'\}$
13:      **end if**
14:      $N' \leftarrow N_1$
15:    **end while**
16: **end for**

---

## C    The Necessity of Eq.(3a) in Program (13)

Now we show the necessity of Eq.(3a) in Program (13). We denote Program **T** as the resulting program after removing Eq.(3a) in Program (13). We use the optimization gap between the optimal objective value $g^*$ in Program (13) and the objective value $\underline{g}$ obtained from the players' strategies after solving Program **T** (i.e., $\underline{g}$ is the real objective value after playing the strategies obtained from solving Program **T**) to measure the inefficiency of Program **T**, i.e., $g^* - \underline{g}$. Note that, by Theorem 2, the optimal objective value of Program **T** is just an upper bound of the optimal objective value of Program (13), which may not be achieved by playing the strategies obtained from solving Program **T**. The following theorem shows that $g^* - \underline{g}$ can be arbitrarily large, i.e., Program **T** is not suitable to be used for computing optimal NEs. In addition, the resulting strategy profile by solving Program **T** may not be an NE.[5]

**Theorem 5.** *$g^* - \underline{g}$ can be arbitrarily large, and the resulting strategy profile $x'$ by solving Program* **T** *may not be an NE.*

*Proof.* Consider a game with three players, $A_1 = \{a_1, a_1'\}$, $A_2 = \{a_2, a_2'\}$ and $A_3 = \{a_3, a_3', a_3''\}$, and the following utility function for three players, respectively, with $k \geq 1$:

$$u = (u_1, u_2, u_3) : (a_1, a_2, a_3) \rightarrow (0.5k, 0.5k, -k);$$
$$u = (u_1, u_2, u_3) : (a_1', a_2', a_3') \rightarrow (0.5k, 0.5k, -k);$$
$$u = (u_1, u_2, u_3) : (a_1, a_2', a_3) \rightarrow (0.125k, 0.125k, -0.25k);$$
$$u = (u_1, u_2, u_3) : (a_1, a_2', a_3') \rightarrow (0.125k, 0.125k, -0.25k);$$
$$u = (u_1, u_2, u_3) : (a_1, a_2', a_3'') \rightarrow (0.1k, 0.15k, -0.25k);$$
$$u = (u_1, u_2, u_3) : \text{other joint actions} \rightarrow (0, 0, 0).$$

The objective function is $g = u_1(x_1, x_2, x_3) + u_2(x_1, x_2, x_3)$. By solving Program **T**, we obtain $x_1'$ with $x_1'(a_1) = 2/3$ and $x_1'(a_1') = 1/3$, $x_2'$ with $x_2'(a_2) = 1/3$ and $x_2'(a_2') = 2/3$, and $x_3'$ with $x_3'(a_3) = 1/2 = x_3'(a_3')$, and $x_3'(a_3'') = 0$. If players 1 and 2 play $x_1'$ and $x_2'$, respectively, $u_3(x_1', x_2', a_3) = -k/3 = u_3(x_1', x_2', a_3')$, and $u_3(x_1', x_2', a_3'') = -k/9$. That is, player 3 will play the pure strategy $a_3''$ to respond to $x_1'$ and $x_2'$, which will result in $\underline{g} = k/9$. Then the resulting strategy profile $x'$ is not an NE.

---

[5]To find an optimal NE, this paper only considers programs guaranteeing exact NEs, and designing programs with approximate NEs is the future work.

It is clear that $(x_1^*, x_2^*, x_3^*)$ with $x_1^*(a_1) = 1$, $x_2^*(a_2') = 1$, and $x_3^*(a_3'') = 1$ is an NE, which also is an output of solving Program (13) with the objective value $g^* = k/4$.

Therefore, $g^* - \underline{g} = 5k/36$, which is arbitrarily large when $k$ is arbitrarily large. $\qquad\square$

---

**Algorithm 4** $Build(N')$: Build a minimum-height binary tree for $N'$

---

1: $h \leftarrow \lceil \log(|N'|) \rceil$
2: **if** $2^h = |N'|$ **then**
3:      Lower set $\mathcal{N}'_1 \leftarrow \{\{i\} \mid i \in N'\}$
4:      **for** $k \in \{1, \ldots, \lceil \log(|\mathcal{N}'|) \rceil\}$ **do**
5:          Upper set $\mathcal{N}'_2 \leftarrow \emptyset$
6:          **for** $j \in \{1, \ldots, |\mathcal{N}'_1|/2\}$ **do**
7:              $N_1 \leftarrow \mathcal{N}'_1[j \times 2 - 1] \cup \mathcal{N}'_1[j \times 2]$: the union of the $(j \times 2 - 1)$-th element and the $(j \times 2)$-th element in $\mathcal{N}'_1$.
8:              $\mathcal{N}'_2 \leftarrow \mathcal{N}'_2 \cup \{N_1\}$
9:              $Ch(N_1) \leftarrow \{\mathcal{N}'_1[j \times 2 - 1], \mathcal{N}'_1[j \times 2]\}$
10:          **end for**
11:          Lower set $\mathcal{N}'_1 \leftarrow \mathcal{N}'_2$
12:          $\underline{\mathcal{N}} \leftarrow \underline{\mathcal{N}} \cup \mathcal{N}'_2$
13:      **end for**
14: **else**
15:      $N'_1 \leftarrow \{N'[1], \ldots, N'[2^{h-1}]\}$
16:      $N'_2 \leftarrow N' \setminus N'_1$
17:      **if** $3 \times 2^{h-2} <= |N'|$ **then**
18:          Lower set $\mathcal{N}'_1 \leftarrow \{\{i\} \mid i \in N'_1\}$
19:          **for** $k \in \{1, \ldots, \lceil \log(|N'_1|) \rceil\}$ **do**
20:              Repeat Lines 5-12.
21:          **end for**
22:          $Build(N'_2)$
23:          $\underline{\mathcal{N}} \leftarrow \underline{\mathcal{N}} \cup \{N'\}$
24:          $Ch(N') \leftarrow \{N'_1, N'_2\}$
25:      **else**
26:          Lower set $\mathcal{N}'_1 \leftarrow \{\{i\} \mid i \in N'_1\}$
27:          **for** $k \in \{1, \ldots, \lceil \log(|N'_1|) \rceil - 1\}$ **do**
28:              Repeat Lines 5-12.
29:          **end for**
30:          $Build(N'_2)$
31:          $N_1 \leftarrow \mathcal{N}'_1[2] \cup N'_2$
32:          $\underline{\mathcal{N}} \leftarrow \underline{\mathcal{N}} \cup \{N_1\}$
33:          $Ch(N_1) \leftarrow \{\mathcal{N}'_1[2], N'_2\}$
34:          $\underline{\mathcal{N}} \leftarrow \underline{\mathcal{N}} \cup \{N'\}$
35:          $Ch(N') \leftarrow \{\mathcal{N}'_1[1], N_1\}$
36:      **end if**
37: **end if**

---

# D    Binary Trees and Details of Algorithm 1

We consider a special binary tree (full binary tree), which includes two kinds of nodes: nodes with two children (internal nodes) and nodes without children (leaf nodes). A binary tree $T_{N'}$ of $N' \subseteq N$ with $|N'| \geq 2$ is that: 1) its root is $N'$; 2) its nodes are $\{N'' \mid N'' \subseteq N'\}$; 3) each of its leaf nodes is a singleton; and 4) each of its internal nodes $N''$ has two children $N''_l$ and $N''_r$ with $N''_l \cap N''_r = \emptyset$ and $N'' = N''_l \cup N''_r$, i.e., $N''$ is divided into two disjoint sets. Let $Ch(N'') = \{N''_l, N''_r\}$ be the set of $N''$'s children in $T_{N'}$, and $Ch(N'') = \emptyset$ if $N''$ is a singleton. Let $\mathcal{N}_{T_{N'}}$ be the set of internal nodes in $T_{N'}$.

Our binary tree for $-i$ is a full binary tree, i.e., each internal node has two children, which has $k - 1$ internal nodes if there are $k$ leaf nodes [20]. For example, Figure 1(a) has 3 internal nodes and 4 leaf

nodes. The length of the path from the root to a leaf is the number of internal nodes on this path in a binary tree. The height of a binary tree is the maximum path length, and the total path length is the sum of the lengths of the paths from the root to each leaf node in a binary tree. For example, the path length from the root to each leaf node in Figure 1(a) is 2, the height is 2, and the total path length is $2 \times 4 = 8$.

To obtain the minimum number of internal nodes in these binary trees for $\{-i \mid i \in N\}$, we can minimize the difference between binary trees. Given the binary tree $T_{-n}$ for $-n$ and the binary tree $T_{-i}$ for $-i$ with $i \in -n$, the difference between $T_{-n}$ and $T_{-i}$ at least includes the path from the root to the leaf node $\{i\}$ in $T_{-n}$ and the path from the root to the leaf node $\{n\}$ in $T_{-i}$. Then the number of different internal nodes (i.e., internal nodes that are not in $T_{-n}$) in these binary trees for $\{-i \mid i \in N\}$ is at least equal to the total path length in $T_{-n}$. Now we propose an algorithm ensuring that the number of different internal nodes in these binary trees for $\{-i \mid i \in N\}$ is equal to the total path length in $T_{-n}$, which is the minimum total path length. To do that, we first build a binary tree for $-n$ with the minimum height (a full binary tree with the minimum height may not be balanced). Note that there are at most $2^h$ leaf nodes in a binary tree with the height $h$, and there are $n - 1$ leaf nodes and $n - 2$ internal nodes in a binary tree for $-n$. Then we can build a full binary tree $T_{-n}$ with the height $\lceil \log_2(n-1) \rceil$ for $-n$ and then replace $i$ with $n$ in the nodes of $T_{-n}$ to obtain $T_{-i}$ for each $i \in -n = \{1, \ldots, n-1\}$. That creates $n$ full binary trees for $\{-i \mid i \in N\}$. This procedure is shown in Algorithm 1, generating our minimum binary collection $\underline{\mathcal{N}}$. Figure 1(a) builds a binary tree $T_{-5}$, and Figure 1(a) obtains $T_{-3}$ by replacing 3 with 5 in $T_{-5}$.

The full details of Algorithm 1 are shown in Algorithm 3. Line 1 builds a binary tree with the height $\lceil \log_2(n-1) \rceil$ for $-n = \{1, \ldots, n-1\}$, whose details are shown in Algorithm 4. At Lines 2-16, for each $i$ in $\{1, \ldots, n-1\}$, we search the binary tree for $-n$ from the root and replace $i$ with $n$ in each node including $i$ to form a new tree for $-i$. And we only need to add new internal nodes to $\underline{\mathcal{N}}$.

Algorithm 4 builds a binary tree with the height $\lceil \log_2(|N'|) \rceil$ for $N'$. If the size of $N'$ is $2^{\lceil \log_2(|N'|) \rceil}$ (note that each element in $N'$ corresponds to a leaf node, and there are at most $2^h$ leaf nodes in a binary tree with the height $h$), then we can build a complete binary tree, where all leaf nodes are at the lowest level. That is, we combine two nodes at the lower level to form a node at the upper level, as shown at Lines 3-12. If the size of $N'$ is not $2^{\lceil \log_2(|N'|) \rceil}$, and it is larger than $3 \times 2^{\lceil \log_2(|N'|) \rceil - 2}$, then we build a complete binary tree for the subset $N'_1$ (Line 15) with the height $\lceil \log_2(|N'_1|) \rceil = \lceil \log_2(|N'|) \rceil - 1$ (Lines 18-20) and then build a binary tree for the remaining subset (Line 22). Finally, we combine both binary trees together to form a binary tree for $N'$ (Line 24). If the size of $N'$ is not $2^{\lceil \log_2(|N'|) \rceil}$, and it is less than $3 \times 2^{\lceil \log_2(|N'|) \rceil - 2}$, we build a complete binary tree for the subset $N'_1$ (Line 28) with the height $\lceil \log_2(|N'_1|) \rceil = \lceil \log_2(|N'|) \rceil - 1$. However, at the last step of building the binary tree for $N'_1$, we do not combine two nodes to form a root. We keep both two nodes within $\mathcal{N}'_1$ by setting $k \leq \lceil \log(|N'_1|) \rceil - 1 = \lceil \log(|N'|) \rceil - 2$ at Line 26. Then we build a binary tree for the remaining subset $N'_2$ (Line 30). After that, we combine the root of the binary tree for $N'_2$ and two nodes in $\mathcal{N}'_1$ to form a binary tree for $N'$ (Line 31-35). This step is to try to reduce the total path length because the number of nodes in the binary tree for $N'_2$ is less than the number of nodes of the binary tree for any node in $\mathcal{N}'_1$, and we can reduce the total path length by combining the root of the binary tree for $N'_2$ and any node in $\mathcal{N}'_1$ to form a node first.

## E   Runtime for Algorithm 1

Table 3 shows the runtime of Algorithm 1, which is extremely small and then can be ignored, compared to the runtime of CRM shown in Tables 4 and 5.

## F   Experiment Setting

**Games:** we evaluate our approach on two sets of games: randomly generated games and games that are generated by GAMUT [31]. Payoffs are generated from the interval between 0 and 100 (other ranges (e.g., $[0, 1]$) do not affect the result). We vary the number of players (i.e., $n$) and the number of actions (i.e., $m$) for each player for random games (i.e., $(n, m)$). For GAMUT games, we use the variants with six players and three actions (i.e., the game $(6, 3)$), which are much larger than the three-player three-action games (i.e., the game $(3, 3)$) used in prior work [4, 13]. We show the game size in terms of the number of bilinear terms and integer variables in Appendix G, e.g., the number of

Table 3: Game size and runtime of Algorithm 1.

| $(n, m)$ | Integer variables Size | Bilinear terms Based on $\overline{\mathcal{N}}$ | Based on $\underline{\mathcal{N}}$ | Correlation Plans $|\overline{\mathcal{N}}|$ | $|\underline{\mathcal{N}}|$ | Algorithm 1 Runtime |
|---|---|---|---|---|---|---|
| (3, 2) | 6 | 12 | 12 | 3 | 3 | <0.0001s |
| (5, 2) | 10 | 200 | 104 | 25 | 11 | 0.0001s |
| (7, 2) | 14 | 2044 | 564 | 119 | 21 | 0.0002s |
| (3, 3) | 9 | 27 | 27 | 3 | 3 | <0.0001s |
| (4, 3) | 12 | 162 | 135 | 10 | 7 | <0.0001s |
| (5, 3) | 15 | 765 | 459 | 25 | 11 | 0.0001s |
| (4, 2) | 8 | 56 | 44 | 10 | 7 | <0.0001s |
| (4, 3) | 12 | 162 | 135 | 10 | 7 | <0.0001s |
| (4, 4) | 16 | 352 | 304 | 10 | 7 | <0.0001s |
| (4, 5) | 20 | 650 | 575 | 10 | 7 | <0.0001s |
| (3, 5) | 15 | 75 | 75 | 3 | 3 | <0.0001s |
| (3, 8) | 24 | 24 | 24 | 3 | 3 | <0.0001s |
| (3, 10) | 30 | 300 | 300 | 3 | 3 | <0.0001s |
| (3, 13) | 39 | 507 | 507 | 3 | 3 | <0.0001s |
| (3, 15) | 45 | 675 | 675 | 3 | 3 | <0.0001s |
| (3, 17) | 51 | 867 | 867 | 3 | 3 | <0.0001s |
| (6, 3) | 18 | 3348 | 1620 | 56 | 16 | 0.0001s |
| (8, 2) | 16 | 6288 | 1172 | 246 | 26 | 0.0003s |
| (9, 2) | 18 | 19152 | 2512 | 501 | 31 | 0.0003s |

Table 4: Results for random games: $(n, m)$ represents the game with $n$ players and $m = |A_i|$ actions for each player. The format for each result is: Average Runtime $\pm$ 95% Confidence Interval (Percentage of Games not Solved within the Time Limit) (Utility Gap). Note that the unit of the runtime is second, and the case that all games have been solved with the time limit should be (0%) and is omitted, we only need to care about the utility gap for EXCLUSION, and the utility gap $\infty$ represents EXCLUSION cannot return a solution within the time limit. For example, for the random games $(7, 2)$, CRM solves 100% of them by using 25s with a 95% interval 17s, but 80% of them are not solved by EXCLUSION within the time limit, and EXCLUSION has a utility gap 53%.

| | | Runtime $\pm$ 95% Confidence Interval (Percentage of Games not Solved) (Utility Gap) | | | |
|---|---|---|---|---|---|
| Vary | $(n, m)$ | CRM | MIBP | ENUMPOLY | EXCLUSION |
| | (3, 2) | **0.01 $\pm$ 0** | 0.02 $\pm$ 0 | 0.03 $\pm$ 0.01 | 31 $\pm$ 41 (gap:15%) |
| $n$ | (5, 2) | **0.2 $\pm$ 0.1** | 0.5 $\pm$ 0.4 | 11 $\pm$ 4 | 753 $\pm$ 148 (73%) (gap:64%) |
| | (7, 2) | **25 $\pm$ 17** | 429 $\pm$ 131 (20%) | 1000 $\pm$ 0 (97%) | 835 $\pm$ 119 (80%) (gap:53%) |
| | (3, 3) | **0.1 $\pm$ 0** | 0.1 $\pm$ 0 | 51 $\pm$ 59 | 252 $\pm$ 140 (20%) (gap:34%) |
| $n$ | (4, 3) | **0.3 $\pm$ 0.1** | 1 $\pm$ 0.3 | 1000 $\pm$ 0 (100%) | 773 $\pm$ 125 (67%) (gap:58%) |
| | (5, 3) | **22 $\pm$ 9** | 239 $\pm$ 87 (7%) | 1000 $\pm$ 0 (100%) | 974 $\pm$ 50 (97%) (gap:62%) |
| | (4, 2) | **0.1 $\pm$ 0.01** | 0.1 $\pm$ 0.01 | 0.2 $\pm$ 0.1 | 246 $\pm$ 126 (13%) (gap:23%) |
| $m$ | (4, 3) | **0.3 $\pm$ 0.1** | 1 $\pm$ 0.3 | 1000 $\pm$ 0 (100%) | 773 $\pm$ 125 (67%) (gap:58%) |
| | (4, 4) | **2.8 $\pm$ 1** | 42 $\pm$ 12 | 1000 $\pm$ 0 (100%) | 1000 $\pm$ 0 (100%) (gap:73%) |
| | (4, 5) | **64 $\pm$ 42** | 862 $\pm$ 91 (77%) | 1000 $\pm$ 0 (100%) | 1000 $\pm$ 0 (100%) (gap:75%) |
| | (3, 5) | **0.2 $\pm$ 0.03** | 0.3 $\pm$ 0.1 | 1000 $\pm$ 0 (100%) | 1000 $\pm$ 0 (100%) (gap:67%) |
| $m$ | (3, 8) | **4 $\pm$ 3** | 247 $\pm$ 140 (17%) | 1000 $\pm$ 0 (100%) | 1000 $\pm$ 0 (100%) (gap:$\infty$) |
| | (3, 10) | **9 $\pm$ 9** | 334 $\pm$ 167 (30%) | 1000 $\pm$ 0 (100%) | 1000 $\pm$ 0 (100%) (gap:$\infty$) |
| | (3, 13) | **38 $\pm$ 21** | 342 $\pm$ 151 (27%) | 1000 $\pm$ 0 (100%) | 1000 $\pm$ 0 (100%) (gap:$\infty$) |

bilinear terms in the game $(9, 2)$ is 19152 based on $\overline{\mathcal{N}}$ but is 2512 based on $\underline{\mathcal{N}}$. For each setting, we generated 30 games on MacBook Pro, where the seeds are $i \in \{1, \ldots, 30\}$ for the GAMUT games and $20201125 + i \cdot 10$ for random games. Results in this section are hence averaged over 30 cases.

**Algorithm Setting:** The objective function used in the experiments maximizes the expected utility of player $n$. We verified that results for optimizing other objectives (e.g., maximizing social welfare) are similar. We use the non-convex solver of Gurobi 9.5 to solve all mixed-integer bilinear programs with the optimality gap set to 0.0001 (the default setting). EXCLUSION uses this optimality gap as well, which is significantly smaller than 0.001 in [4] (we verified that, with the same optimality gap, our result for EXCLUSION is almost the same as the one in [4]). Experiments are run on an eight-core Intel Core I9 machine at 2.3 GHz with 16GB of RAM. Similarly to the previous literature [34, 4, 13],

Table 5: Results for six-player three-action GAMUT games.

| Game | Runtime ± 95% Confidence Interval (Percentage of Games not Solved) (Utility Gap) | | | |
|---|---|---|---|---|
| | CRM | MIBP | ENUMPOLY | EXCLUSION |
| Bidirectional LEG | **1.6** ± **1** | 972 ± 54 (97%) | 1000 ± 0 (100%) | 1000 ± 0 (100%) (gap:13%) |
| Collaboration | **1** ± **0.2** | 967 ± 63 (97%) | 1000 ± 0 (100%) | 1000 ± 0 (100%) (gap:81%) |
| Covariant $r = 0.5$ | **5** ± **6** | 1000 ± 0 (100%) | 1000 ± 0 (100%) | 963 ± 59 (93%) (gap:73%) |
| PolyMatrix | **26** ± **44** | 194 ± 74 (3%) | 867 ± 116 (87%) | 1000 ± 0 (100%) (gap:17%) |
| Random LEG | **2** ± **1** | 1000 ± 0 (100%) | 1000 ± 0 (100%) | 986 ± 27 (97%) (gap:11%) |
| Random graphical | **0.1** ± **0.1** | 803 ± 140(83%) | 50 ± 30 | 971 ± 55 (97%) (gap:32%) |
| Uniform LEG | **2.2** ± **1** | 1000 ± 0 (100%) | 1000 ± 0 (100) | 986 ± 26 (97%) (gap:11%) |

Table 6: Ablation study. No time limit for CRM, CR, and CM in games (8, 2). We can see that each component of our approach significantly boosts its performance. Note that $\overline{\mathcal{N}}$ (in CR, C, and MIBP) and $\underline{\mathcal{N}}$ (in CRM, CM, and M) result in the same bilinear terms in three-player games because each element in $\{-i \mid i \in \{1, 2, 3\}\}$ includes only two elements such that Algorithm 1 cannot reduce the number of internal nodes to reduce the number of bilinear terms, where CR and CRM (or C and CM, or MIBP and M) have the same performance. For one case in the game (9, 2), which CRM and CM cannot solve within 1000s, after removing the time limit for it, CRM solves it by using 1198s, but CM solves it by using 12751s. The unit of the runtime is second.

| Game | Runtime ± 95% Confidence Interval (Percentage of Games not Solved) | | | | | |
|---|---|---|---|---|---|---|
| | CRM | CR | CM | C | M | MIBP |
| (9, 2) | **658±128 (50%)** | 988 ± 23 (97%) | 782±113 (63%) | 1000± 0 (100%) | 1000 ± 0 (100%) | 1000 ± 0 (100%) |
| (8, 2) | **166± 97** | 2334± 1742 | 278± 207 | 763 ± 120 (60%) | 1000 ± 0 (100%) | 1000 ± 0 (100%) |
| (7, 2) | **25 ± 17** | 89 ± 51 | 36 ± 28 | 408 ± 157 (30%) | 488 ± 111 (10%) | 429 ± 131 (20%) |
| (3, 15) | **167± 86 (3%)** | 167 ± 86 (3%) | 317± 137 (17%) | 317 ± 137 (17%) | 558 ± 150 (40%) | 558 ± 150 (40%) |
| (3, 17) | **231±122 (10%)** | 231 ±122 (10%) | 326± 134 (20%) | 326 ± 134 (20%) | 784 ± 102 (53%) | 784 ± 102 (53%) |
| Bidirectional LEG | **1.6 ± 1** | 5.4 ± 4 | 2.2 ± 2 | 86 ± 3 | 991 ± 18 (97%) | 972 ± 54 (97%) |
| Collaboration | **1 ± 0.2** | 2 ± 0.2 | 1 ± 0.1 | 2 ± 0.4 | 867 ± 122 (87%) | 967 ± 63 (97%) |
| Covariant $r = 0.5$ | **5 ± 6** | 12 ± 10 | 5 ± 6 | 18 ± 18 | 1000 ± 0 (100%) | 1000 ± 0 (100%) |
| PolyMatrix | 26 ± 44 | 33 ± 25 | 36 ± 64 (3%) | **17 ± 21** | 267 ± 97 (10%) | 194 ± 74 (3%) |
| Random LEG | **2 ± 1** | 6 ± 5 | 2.5 ± 2 | 5 ± 5 | 1000 ± 0 (100%) | 1000 ± 0 (100%) |
| Random graphical | **0.1 ± 0.1** | 0.4 ± 0.1 | 0.2 ± 0.1 | 0.6 ± 0.4 | 814 ± 134(80%) | 803 ± 140(83%) |
| Uniform LEG | **2.2 ± 1** | 5 ± 4 | 2.5 ± 2 | 5 ± 5 | 999 ± 2 (97%) | 1000 ± 0 (100%) |

to efficiently evaluate the algorithms, we set a time limit of 1000 seconds for each case unless stated otherwise.

**Metric:** We use the runtime and the percentage of games that are not solved within the time limit to measure the performance of our approach. In addition to the average runtime, we show a 95% confidence interval. CRM and MIBP guarantee finding an optimal NE. For algorithms that can enumerate all NEs, we can choose an optimal NE from the output for all NEs. We then compare ENUMPOLY to our approach only in the runtime. For algorithms that only guarantee to converge to an NE, we may need to use them to enumerate all NEs. However, it is unclear whether algorithms like EXCLUSION can enumerate all NEs. Therefore, we compare EXCLUSION to our approach in the runtime and the *utility gap*. The utility gap is the relative distance between the optimal objective value ($g^*$) in our problem (returned by CRM) and the objective value ($g_0$) in the Nash equilibrium returned by EXCLUSION, i.e., $|g^* - g_0|/|g_0| \times 100\%$. In some cases, a solution is returned even if it has not reached the given accuracy within the time limit, which is still used as a solution of EXCLUSION. A larger gap means that we will lose more while using EXCLUSION.

# G   Game Size

Table 3 shows the number of integer variables, bilinear terms, and correlation plans for the games we used in experiments. Note that the used GAMUT games have six players and three actions. Also, note that we do not reduce the number of bilinear terms and correlation plans in games with only three players.

# H   Details of Experimental Results

The details of experimental results are in Tables 4, 5, and 6.

Specially, CR is solving Program (13) based on $\overline{\mathcal{N}}$, and CRM is solving Program (13) based on $\underline{\mathcal{N}}$. We discussed the difference between $\overline{\mathcal{N}}$ and $\underline{\mathcal{N}}$ in Section 3.3. Basically, $\overline{\mathcal{N}}$ and $\underline{\mathcal{N}}$ are the same in

Table 7: Results on more Gambit algorithms for random games. The format for each result is: Average Runtime ± 95% Confidence Interval (Percentage of Games not Solved within the Time Limit). Note that, these Gambit algorithms only achieve some NE if the game is solved, which may not be optimal. Even so, these algorithms fail to solve many games, and even run significantly slower than our CRM (see Table 4) in many games.

| $(n, m)$ | Runtime ± 95% Confidence Interval (Percentage of Games not Solved) | | | | |
|---|---|---|---|---|---|
| | GNM | IPA | LIAP | SIMPDIV | LOGIT |
| (3, 2) | 0.03 ± 0.02 | 567 ± 177 (57%) | 0.06 ± 0.02 (77%) | 0.07 ± 0.06 | 0.04 ± 0.02 (100%) |
| (5, 2) | 0.04 ± 0.01 (3%) | 867 ± 122 (87%) | 0.45 ± 0.04 (100%) | 1000 ± 0 (100%) | 0.02 ± 0 (100%) |
| (7, 2) | 333 ± 169 (53%) | 400 ± 175 (37%) | 6 ± 0.4 (100%) | 79.4 ± 78.6 | 0.05 ± 0.05 (100%) |
| (3, 3) | 0.03 ± 0 (3%) | 933 ± 89 (93%) | 0.16 ± 0.02 (100%) | 300 ± 163 (30%) | 0.04 ± 0.02 (100%) |
| (4, 3) | 0.17 ± 0.11 (3%) | 500 ± 179 (50%) | 0.76 ± 0.08 (100%) | 500 ± 179 (50%) | 0.02 ± 0 (100%) |
| (5, 3) | 0.15 ± 0.03 (30%) | 773 ± 158 (73%) | 6.3 ± 0.3 (100%) | 900 ± 107 (90%) | 0.02 ± 0.01 (100%) |
| (4, 2) | 0.03 ± 0.01 (3%) | 667 ± 169 (63%) | 0.13 ± 0.02 (97%) | 733 ± 158 (73%) | 0.02 ± 0 (100%) |
| (4, 3) | 0.17 ± 0.11 (3%) | 500 ± 179 (50%) | 0.76 ± 0.08 (100%) | 500 ± 179 (50%) | 0.02 ± 0 (100%) |
| (4, 4) | 800 ± 143 (80%) | 367 ± 172 (33%) | 4.9 ± 0.4 (100%) | 867 ± 121 (87%) | 0.07 ± 0.07 (100%) |
| (4, 5) | 0.33 ± 0.07 (37%) | 833 ± 133 (83%) | 16 ± 1 (100%) | 900 ± 107 (90%) | 0.04 ± 0.03 (100%) |
| (3, 5) | 600 ± 175 (63%) | 767 ± 151 (77%) | 1.5 ± 0.1 (100%) | 933 ± 89 (93%) | 0.06 ± 0.09 (100%) |
| (3, 8) | 0.76 ± 0.26 (17%) | 506 ± 177 (50%) | 11 ± 0.7 (100%) | 867 ± 122 (87%) | 0.02 ± 0 (100%) |
| (3, 10) | 334 ± 168 (53%) | 767 ± 153 (77%) | 37 ± 3 (100%) | 867 ± 121 (87%) | 0.02 ± 0 (100%) |
| (3, 13) | 3.8 ± 1.1 (47%) | 1000 ± 0 (100%) | 132 ± 11 (100%) | 805 ± 140 (80%) | 0.03 ± 0 (100%) |

Table 8: Results on more Gambit algorithms for six-player three-action GAMUT games. The format for each result is: Average Runtime ± 95% Confidence Interval (Percentage of Games not Solved within the Time Limit). Note that, these Gambit algorithms only achieve some NE if the game is solved, which may not be optimal. Even so, these algorithms fail to solve many games, and even run significantly slower than our CRM (see Table 5) in many games.

| Game | Runtime ± 95% Confidence Interval (Percentage of Games not Solved) | | | | |
|---|---|---|---|---|---|
| | GNM | IPA | LIAP | SIMPDIV | LOGIT |
| Bidirectional LEG | 167 ± 133 (63%) | 10 ± 17 | 8 ± 0.6 (100%) | 667 ± 169 (67%) | 0.03 ± 0 (100%) |
| Collaboration | 34 ± 64 (3%) | 0.03 ± 0 | 24 ± 2 (100%) | 0.03 ± 0 | 0.03 ± 0 (100%) |
| Covariant $r = 0.5$ | 100 ± 107 (17%) | 0.04 ± 0 | 21 ± 2 (100%) | 367 ± 172 (37%) | 0.04 ± 0.03 (100%) |
| PolyMatrix | 0.13 ± 0.03 (7%) | 8.3 ± 8 | 9 ± 0.8 (100%) | 24 ± 27 | 0.05 ± 0.05 (100%) |
| Random LEG | 0.19 ± 0.04 (47%) | 0.04 ± 0 | 7.5 ± 0.6 (100%) | 777 ± 146 (77%) | 0.03 ± 0 (100%) |
| Random graphical | 0.05 ± 0 (3%) | 1000 ± 0 (100%) | 9 ± 1 (100%) | 0.05 ± 0.03 | 0.04 ± 0.02 (100%) |
| Uniform LEG | 0.16 ± 0.04 (47%) | 0.04 ± 0 | 7.3 ± 0.6 (100%) | 776 ± 146 (77%) | 0.02 ± 0 (100%) |

3-player games, so CR and CRM have the same performance in 3-player games shown in Table 6. When the number of players increases, the size of $\overline{\mathcal{N}}$ is larger and larger than $\underline{\mathcal{N}}$, and then CRM's advantage over CR is more significant. This statement is verified by our result: game (7, 2): CRM with 25 ± 17 and CR with 89 ± 51; and game (8, 2): CRM with 156 ± 83 (3%) and CR with 612 ± 129 (33%) (see Table 2). However, from the game (7, 2) to the game (8, 2), the trend that CRM's advantage over CR is more significant is not very clear based on the above data due to the time limit in the game (8, 2). To show this trend clearly, we remove the time limit for CRM and CR in the game (8, 2) and obtain: CRM with 166 ± 97 and CR with 2334 ± 1742, which is shown in Table 6. Then from the game (7, 2), where CRM is about 3 times faster than CR, to the game (8, 2), where CRM is about 13 times faster than CR, we can clearly see the trend that CRM's advantage over CR is more significant when the number of players increases. The difference between $\overline{\mathcal{N}}$ and $\underline{\mathcal{N}}$ also explains that CR is slower than CM in the game (8, 2).

# I   Results on More Gambit Algorithms

Results in Tables 7 and 8 show that Gambit algorithms, i.e., GNM, IPA, LIAP, SIMPDIV, and LOGIT, cannot guarantee finding an NE, i.e., fail to solve many games, and even run significantly slower than our CRM (see Tables 4 and 5) in many games:

- GNM fails to solve many large games: GNM stops without output in some of these cases, and cannot stop within the time limit in other cases. GNM runs significantly slower than our CRM in many games, e.g., random games $(7, 2), (4, 4), (3, 5), (3, 10)$, and GAMUT games Bidirecttional LEG, Collaboration, Covariant.

- IPA fails to solve many large games: IPA cannot stop within the time limit in most of these games. IPA runs significantly slower than our CRM in most random games, and the GAMUT game Random graphical.

- LIAP can only solve several games in small random games $(3, 2)$ and $(4, 2)$, and fails to solve all of the other games: LIAP stops without output in these games. Even so, LIAP runs significantly slower than our CRM in most games.

- SIMPDIV fails to solve many large games: SIMPDIV cannot stop within the time limit in almost all of these games. SIMPDIV runs significantly slower than our CRM in most games.

- LOGIT fails to solve all of these games: LOGIT stops without output in all of these games. LOGIT may not work properly in this latest GAMBIT version. Even so, LOGIT runs significantly slower than our CRM in the random game $(3, 2)$.

