# Computing Optimal Nash Equilibria in Multiplayer Games

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

 | **1.6** $\pm$ **1** | 972 $\pm$ 54 (97%) | 1000 $\pm$ 0 (100%) | 1000 $\pm$ 0 (100%) (gap:13%) |
| Collaboration | **1** $\pm$ **0.2** | 967 $\pm$ 63 (97%) | 1000 $\pm$ 0 (100%) | 1000 $\pm$ 0 (100%) (gap:81%) |
| Covariant $r = 0.5$ | **5** $\pm$ **6** | 1000 $\pm$ 0 (100%) | 1000 $\pm$ 0 (100%) | 963 $\pm$ 59 (93%) (gap:73%) |
| PolyMatrix | **26** $\pm$ **44** | 194 $\pm$ 74 (3%) | 867 $\pm$ 116 (87%) | 1000 $\pm$ 0 (100%) (gap:17%) |
| Random LEG | **2** $\pm$ **1** | 1000 $\pm$ 0 (100%) | 1000 $\pm$ 0 (100%) | 986 $\pm$ 27 (97%) (gap:11%) |
| Random graphical | **0.1** $\pm$ **0.1** | 803 $\pm$ 140(83%) | 50 $\pm$ 30 | 971 $\pm$ 55 (97%) (gap:32%) |
| Uniform LEG | **2.2** $\pm$ **1** | 1000 $\pm$ 0 (100%) | 1000 $\pm$ 0 (100) | 986 $\pm$ 26 (97%) (gap:11%) |

## E Runtime for Algorithm 1

Table 3 shows the runtime of Algorithm 1, which is extremely small and then can be ignored, compared to the runtime of CRM shown in Tables 4 and 5.

## F Experiment Setting

**Games:** we evaluate our approach on two sets of games: randomly generated games and games that are generated by GAMUT [29]. Payoffs are generated from the interval between 0 and 100 (other ranges (e.g., $[0, 1]$) do not affect the result). We vary the number of players (i.e., $n$) and the number of actions (i.e., $m$) for each player for random games (i.e., $(n, m)$). For GAMUT games, we use the variants with six players and three actions (i.e., the game $(6, 3)$), which are much larger than the three-player three-action games (i.e., the game $(3, 3)$) used in prior work [4, 13]. We show the game size in terms of the number of bilinear terms and integer variables in Appendix G, e.g., the number of bilinear terms in the game $(9, 2)$ is 19152 based on $\overline{\mathcal{N}}$ but is 2512 based on $\mathcal{N}$. For each setting, we generated 30 games, where the seeds are $i \in \{1, \ldots, 30\}$ for the GAMUT games and $20201125 + i \cdot 10$ for random games. Results in this section are hence averaged over 30 cases.

**Algorithm Setting:** The objective function used in the experiments maximizes the expected utility of player $n$. We verified that results for optimizing other objectives (e.g., maximizing social welfare) are similar. We use the non-convex solver of Gurobi 9.5 to solve all mixed-integer bilinear programs with the optimality gap set to 0.0001 (the default setting). EXCLUSION uses this optimality gap as well, which is significantly smaller than 0.001 in [4] (we verified that, with the same optimality gap, our

Table 6: Ablation study. No time limit for CRM, CR, and CM in games (8, 2). We can see that each component of our approach significantly boosts its performance. Note that $\overline{\mathcal{N}}$ (in CR, C, and MIBP) and $\underline{\mathcal{N}}$ (in CRM, CM, and M) result in the same bilinear terms in three-player games because each element in $\{-i \mid i \in \{1, 2, 3\}\}$ includes only two elements such that Algorithm 1 cannot reduce the number of internal nodes to reduce the number of bilinear terms, where CR and CRM (or C and CM, or MIBP and M) have the same performance. For one case in the game (9, 2), which CRM and CM cannot solve within 1000s, after removing the time limit for it, CRM solves it by using 1198s, but CM solves it by using 12751s. The unit of the runtime is second.

| Game | Runtime $\pm$ 95% Confidence Interval (Percentage of Games not Solved) | | | | | |
|---|---|---|---|---|---|---|
|  | CRM | CR | CM | C | M | MIBP |
| (9, 2) | **658±128 (50%)** | 988 $\pm$ 23 (97%) | 782±113 (63%) | 1000± 0 (100%) | 1000 $\pm$ 0 (100%) | 1000 $\pm$ 0 (100%) |
| (8, 2) | **166± 97** | 2334± 1742 | 278± 207 | 763 $\pm$ 120 (60%) | 1000 $\pm$ 0 (100%) | 1000 $\pm$ 0 (100%) |
| (7, 2) | **25 $\pm$ 17** | 89 $\pm$ 51 | 36 $\pm$ 28 | 408 $\pm$ 157 (30%) | 488 $\pm$ 111 (10%) | 429 $\pm$ 131 (20%) |
| (3, 15) | **167± 86 (3%)** | 167 $\pm$ 86 (3%) | 317± 137 (17%) | 317 $\pm$ 137 (17%) | 558 $\pm$ 150 (40%) | 558 $\pm$ 150 (40%) |
| (3, 17) | **231±122 (10%)** | 231 ±122 (10%) | 326± 134 (20%) | 326 $\pm$ 134 (20%) | 784 $\pm$ 102 (53%) | 784 $\pm$ 102 (53%) |
| Bidirectional LEG | **1.6 $\pm$ 1** | 5.4 $\pm$ 4 | 2.2 $\pm$ 2 | 86 $\pm$ 3 | 991 $\pm$ 18 (97%) | 972 $\pm$ 54 (97%) |
| Collaboration | **1 $\pm$ 0.2** | 2 $\pm$ 0.2 | 1 $\pm$ 0.1 | 2 $\pm$ 0.4 | 867 $\pm$ 122 (87%) | 967 $\pm$ 63 (97%) |
| Covariant $r = 0.5$ | **5 $\pm$ 6** | 12 $\pm$ 6 | 5 $\pm$ 6 | 18 $\pm$ 18 | 1000 $\pm$ 0 (100%) | 1000 $\pm$ 0 (100%) |
| PolyMatrix | 26 $\pm$ 44 | 33 $\pm$ 25 | 36 $\pm$ 64 (3%) | **17 $\pm$ 21** | 267 $\pm$ 97 (10%) | 194 $\pm$ 74 (3%) |