# OpenReview forum: "Computing Optimal Nash Equilibria in Multiplayer Games"
_NeurIPS.cc/2023/Conference — NeurIPS 2023 poster_

### Official Review · Reviewer_oqCe · 2023-07-03

**Soundness:** 3 good
**Presentation:** 2 fair
**Contribution:** 2 fair
**Rating:** 5
**Confidence:** 4

**Summary:**

This paper addresses the problem of computing optimal Nash Equilibria in multi-player games. The authors present an optimization algorithm that uses a correlation plan-based formulation with a suitable convex relaxation in order to compute an optimal NE. At its core, the algorithm relies on a binary-tree structure of the correlation plans that enables the adoption of bilinear constraints to represent the set of NE. Finally, the authors experimentally evaluate their algorithm against different baselines for computing optimal NE.

**Strengths:**

The main strength of this work is that the experimental evaluation shows promising results with respect to the state-of-the-art algorithms evaluated by the authors.
Moreover, for what concerns me, the theoretical claims are sound.

**Weaknesses:**

The aspect of this paper that concerns me the most is the contribution and significance of the presented results. I do not find it surprising that you can use correlation plans to compute a NE, and the algorithm formulation seems to me quite straightforward.
Furthermore, while the experimental evaluation shows that in the benchmarks used, the algorithm outperforms existent baselines, I find the games used as benchmarks quite small. This, in my opinion, contributes to limiting the significance of the present paper.

Overall, I don't think that at its current state, this work is suitable for a venue like NeurIPS

**Questions:**

I would like the authors to address my concerns on the significance of the paper.

**Limitations:**

Yes

---

> ### Author Rebuttal · Authors · 2023-08-09
>
> Thank you very much for your comments.
>
> **Q**: About the contribution, significance, and small benchmarks
>
> **Answer**: 1. To our knowledge, as we discussed in the related work section, the approach of breaking the strategy space into bilinear correlation plans based on a binary-tree structure and then heuristically reducing the feasible space, as a path to computing optimal NEs using bilinear programming is novel. We believe the heuristics that are applied over this structure are also good original contributions: There are no existing works that compute optimal NEs in multiplayer games by exploiting correlation plans with their relations based on a binary-tree structure to strictly reduce the feasible solution space after the convex relaxation while minimizing the number of correlation plans to reduce the number of bilinear terms.\
> 2. Based on our general transformation framework, the straightforward approach is using the vanilla binary collection, but our CRM can generate a minimum binary collection, which dramatically reduces time complexity, i.e., getting rid of a term $2^n$ for a term n log n. The improvement on the term $2^n$ in complexity is significant because, theoretically, it means that our proposed algorithm is significantly faster than the straightforward one. The reason is that our proposed algorithm requires significantly fewer bilinear terms than the straightforward one.\
> 3. Our experiment shows that our proposed algorithm is significantly faster than the baselines. For example, for solving the GAMUT game called Random graphical shown in Table 1, our algorithm (i.e., CRM) used about 0.1 seconds, but the straightforward algorithm (i.e., MIBP) used more than 800 seconds.\
> 4. As we mentioned in the limitation section, we cannot handle extremely large games now because we are handling a very hard problem, and then it is unrealistic to expect that our exact algorithm CRM could run very fast in large games. Our algorithm is an attempt to make this computation of optimal NEs feasible. \
> 5. Our algorithmic framework can be built on by further innovative heuristics to improve the computation of optimal NEs. For example, as shown in our additional experiment results (see our response to Reviewer PoVn), our algorithm CRM can solve large-scale real-world games with the aid of the PSRO framework, even when we cannot enumerate all actions due to the memory constraint.

---

> > ### Comment · Reviewer_oqCe · 2023-08-16
> >
> > Thank you for the response.
> >
> > After the rebuttal, I am more positive about the contribution brought by the paper.

---

> > > ### Author Response · Authors · 2023-08-17
> > >
> > > Thank you very much for reviewing our response and raising the score. We welcome new comments if you have any remaining uncertainties.

---

### Official Review · Reviewer_CmiG · 2023-07-05

**Soundness:** 3 good
**Presentation:** 3 good
**Contribution:** 3 good
**Rating:** 7
**Confidence:** 3

**Summary:**

This papers studies computing Nash equilibrium in multiplayer games that optimizes a given objective function. The designed solving framework first transform the corresponding multilinear program into a bilinear program by introducing auxiliary variables representing probability distribution over players' joint actions. Then the feasible solution space after the convex relaxation of bilinear terms can be reduced. Time complexity of the proposed algorithm is shown to be reduced then SOTA both theoretically and numerically.

**Strengths:**

1. The design of correlation plan to reduce the feasible solution space after relaxation is interesting.
2. The proposed approach is shown to be effectively by both concrete thereotical proofs and numerically experiments.

**Weaknesses:**

1. Although the time complexity of solving the optimal Nash is reduced, it is stiil exponential-time, which is not surprised.
2. The notations related to the correlation plan is complicated. Although the examples provide quite clear intuition, it would be better there is any way to simplify the notations.

**Questions:**

NA

---

> ### Author Rebuttal · Authors · 2023-08-09
>
> Thank you very much for your helpful comments.
>
> **Q**: About “Although the time complexity of solving the optimal Nash is reduced, it is stiil exponential-time, which is not surprised.”\
> **Answer**: Based on our general transformation framework, the straightforward approach is using the vanilla binary collection, but our CRM can generate a minimum binary collection to reduce time complexity. The improvement on the term $2^n$ is crucial because:\
> 1.	Theoretically, it means that our proposed algorithm is significantly faster than the straightforward one. The reason is that our proposed algorithm requires significantly fewer bilinear terms than the straightforward one.\
> 2.	Experimentally, our proposed algorithm is significantly faster than the baselines including the straightforward one, which validates our theoretical results. For example, for solving the GAMUT game called Random graphical shown in Table 1, our algorithm (i.e., CRM) used about 0.1 seconds, but the straightforward algorithm (i.e., MIBP) used more than 800 seconds. \
> 3.	As we mentioned in the limitation section, we are handling a very hard problem, and it is unrealistic to expect that we could remove all exponential terms to obtain an extremely fast algorithm. Our algorithm is an attempt to make the computation of optimal NEs feasible, and our algorithmic framework can be built on by further innovative heuristics to improve the computation of optimal NEs. For example, as shown in our additional experiment results (see our response to Reviewer PoVn), our algorithm CRM can solve large-scale real-world games with the aid of the PSRO framework, even when we cannot enumerate all actions due to the memory constraint.

---

> > ### Comment · Reviewer_CmiG · 2023-08-17
> >
> > Thanks for your response.

---

> > > ### Author Response · Authors · 2023-08-17
> > >
> > > Thank you once again for your positive review.

---

### Official Review · Reviewer_PoVn · 2023-07-06

**Soundness:** 4 excellent
**Presentation:** 4 excellent
**Contribution:** 4 excellent
**Rating:** 7
**Confidence:** 3

**Summary:**

This paper tackles the challenge of computing a NE that optimizes some objective (e.g, social welfare). They present an algorithm that avoids the naive exponential blowup that occurs when trying to extend the two-player MILP for optimal NE to multiplayer optimal NE, by using relations between correlation plans to prune the feasible space that results after convex relaxation and reduce the number of bilinear terms. They present empirical evidence of the benefit of this approach.

**Strengths:**

1. The paper is well-written and the contributions made easy to understand.
2. The paper presents a state-of-the-art algorithm for computation of optimal NE in multiplayer games. This is an important problem to consider because it helps operationalize game theory in the real world (most multi-agent settings are not two-player zero-sum).
3. The paper contributes to a growing literature on cooperative AI (e.g., it might be useful to be able to compute an optimal equilibrium so that agents may be steered towards it).

**Weaknesses:**

1.	Would be good to have shown a couple experiments using CRM as the meta-solver for PSRO (as the authors themselves suggest), and maybe compare using other meta-solvers for PSRO, just to see how the performances compared in those settings.

**Questions:**

As mentioned above, it would be good to have some experiments combining this algorithm with other techniques to see how it can be used to solve larger scale games.

It is slightly confusing to me the use of “left” and “right” in line 126 to order the children in some sense of the binary collection. While it is common to refer to “left child” and “right child” of a binary tree, it doesn’t seem especially relevant (could you point out where the ordering of “left” and “right” gets used) to future discussion.

I have read the rebuttal and my concerns have been sufficiently addressed.

**Limitations:**

Yes, the limitations have been addressed.

---

> ### Author Rebuttal · Authors · 2023-08-09
>
> Thank you very much for your valuable comments.
>
> **Q**: About experiments combining this algorithm with other techniques to see how it can be used to solve larger scale games.\
> **Answer**: We conducted experiments on real-world network security games (Jain et al., 2011). In these games, the attacker begins at a source node and traverses by choosing path to one of his targets. The action space of the attacker thus consists of all possible paths. The police officers move independently, each occupying one of the edges of the network in an attempt to apprehend the attacker before he reaches his target. There are three players in these games. The edges of the network with L x W nodes are randomly generated. It is estimated that in a fully connected network with 20 nodes and 190 edges, the number of possible attacker paths is approximately $6.6^{18}$ (Jain et al., 2011).
>
> Our experimental results show that: in games on the network with the 6 x 6 nodes, CRM fails to output the result as it is running out of memory, but our CRM with the aid of the PSRO framework (CRM is used as the meta solver) can solve these games within about 10 seconds. In significantly larger games on the network with 10 x 10 nodes, our CRM, with the aid of the PSRO framework, can solve these games within about 100 seconds. Therefore, our CRM can solve large-scale real-world games with the aid of the PSRO framework, even when we cannot enumerate all actions due to the memory constraint.
>
> Jain, M., Korzhyk, D., Vaněk, O., Conitzer, V., Pěchouček, M. and Tambe, M., 2011, May. A double oracle algorithm for zero-sum security games on graphs. In The 10th International Conference on Autonomous Agents and Multiagent Systems-Volume 1 (pp. 327-334).
>
> **Q**: About the usage of the ordering of “left” and “right”\
> **Answer**: In Line 126, we mentioned: “Let $N'_l$ and $N'_r$ be the left child and the right child of $N'\in \mathcal{N}$, respectively”, which means that each element $N'$ in a binary collection $\mathcal{N}$ has the binary division, i.e., it is divided into two disjoint sets $N'_l$ and $N'_r$. Based on this binary division, any joint action can be divided into two sub-joint actions, as mentioned in Line 132. Therefore, it is used when we divide any joint action into two sub-joint actions, e.g., in Example 1 and Eq.(3a).

---

> > ### Comment · Reviewer_PoVn · 2023-08-14
> >
> > Thank you for your response.

---

> > > ### Author Response · Authors · 2023-08-17
> > >
> > > Thank you once again for your positive review.

---

### Official Review · Reviewer_WEGb · 2023-07-07

**Soundness:** 3 good
**Presentation:** 3 good
**Contribution:** 2 fair
**Rating:** 5
**Confidence:** 3

**Summary:**

The paper attempts an improvement to existing approaches to tackling the problem of optimal Nash equilibrium (NE) computation. In general, the problem is NP-hard.

Usually, a common approach in computing the optimal NE is to formulate it as the solution of a constrained mathematical program whose objective function assesses the optimality of a given NE while the feasibility set describes the set of all NE's. The solution space is nonconvex; commonly, the constraint set is relaxed to a convex one based on the set of correlation plans. The authors devise a a way to shrink the size of the underlying trees and shave-off a term that is exponential to the number of players $n$ from the running time complexity of solving a mixed integer bilinear program.

**Strengths:**

* The analysis seems self-contained, and the previous work is well demonstrated.
* The paper is complemented with a good chunk of experiments.

**Weaknesses:**

* The improvement in the complexity manages to get rid of a term $2^n$ for a term $n \log n$. Still though, in both cases, these terms are multiplied with a term $m^n$ (the number of actions $m$ to the power of the number $n$ of agents). For a game to be nontrivial, $m>1$.


**Questions:**

* Is the improvement on the term $2^n$ that crucial?

**Limitations:**

The limitations are adequately discussed in my opinion.

---

> ### Author Rebuttal · Authors · 2023-08-09
>
> Thank you very much for your comments.
>
> **Q**: Is the improvement on the term $2^n$ that crucial?\
> **Answer**: Yes. Based on our general transformation framework, the straightforward approach is using the vanilla binary collection, but our CRM can generate a minimum binary collection to reduce time complexity. The improvement on the term $2^n$ is crucial because:\
> 1.	Theoretically, it means that our proposed algorithm is significantly faster than the straightforward one. The reason is that our proposed algorithm requires significantly fewer bilinear terms than the straightforward one.\
> 2.	Experimentally, our proposed algorithm is significantly faster than the baselines including the straightforward one, which validates our theoretical results. For example, for solving the GAMUT game called Random graphical shown in Table 1, our algorithm (i.e., CRM) used about 0.1 seconds, but the straightforward algorithm (i.e., MIBP) used more than 800 seconds. \
> 3.	As we mentioned in the limitation section, we are handling a very hard problem, and it is unrealistic to expect that we could remove all exponential terms to obtain an extremely fast algorithm. Our algorithm is an attempt to make the computation of optimal NEs feasible, and our algorithmic framework can be built on by further innovative heuristics to improve the computation of optimal NEs. For example, as shown in our additional experiment results (see our response to Reviewer PoVn), our algorithm CRM can solve large-scale real-world games with the aid of the PSRO framework, even when we cannot enumerate all actions due to the memory constraint.

---

> > ### Comment · Reviewer_WEGb · 2023-08-16
> > **Thank you**
> >
> > I thank the authors for their response. I would like to raise my score to 5 as I am more positive after the rebuttal.

---

> > > ### Author Response · Authors · 2023-08-17
> > >
> > > Thank you very much for reviewing our response and raising the score. We welcome new comments if you have any remaining uncertainties.

---

### Decision · Program_Chairs · 2023-09-21

**Decision:**

Accept (poster)

**Comment:**

Clear accept with unanimous support from all reviewers. The authors are encouraged to address the comments of the reviewers as well as include clarifications emerged during the discussion phase in improving the final version of their paper.